# MUSTARD: MASTERING UNIFORM SYNTHESIS OF THEOREM AND PROOF DATA

**Yinya Huang**[1,6]   **Xiaohan Lin**[2]   **Zhengying Liu**[3†]   **Qingxing Cao**[2†]   **Huajian Xin**[2]
**Haiming Wang**[2]   **Zhenguo Li**[3]   **Linqi Song**[1,6†]   **Xiaodan Liang**[2,4,5†]
[1]City University of Hong Kong   [2]Shenzhen Campus of Sun Yat-sen University
[3]Huawei Noah's Ark Lab   [4]DarkMatter AI Research   [5]MBZUAI   [6]CityUSRI
yinya.huang@hotmail.com, linxh55@mail2.sysu.edu.cn,
{liuzhengying2, Li.Zhenguo}@huawei.com, caoqx8@sysu.edu.cn,
linqi.song@cityu.edu.hk, xdliang328@gmail.com

## ABSTRACT

Recent large language models (LLMs) have witnessed significant advancement in various tasks, including mathematical reasoning and theorem proving. As these two tasks require strict and formal multi-step inference, they are appealing domains for exploring the reasoning ability of LLMs but still face important challenges. Previous studies such as Chain-of-Thought (CoT) have revealed the effectiveness of intermediate steps guidance. However, such step-wise annotation requires heavy labor, leading to insufficient training steps for current benchmarks. To fill this gap, this work introduces MUSTARD, a data generation framework that **m**asters **u**niform **s**ynthesis of **t**heorem **a**nd p**r**oof **d**ata of high quality and diversity. MUSTARD synthesizes data in three stages: (1) It samples a few mathematical concept seeds as the problem category. (2) Then, it prompts a generative language model with the sampled concepts to obtain both the problems and their step-wise formal solutions. (3) Lastly, the framework utilizes a proof assistant (e.g., Lean Prover) to filter the valid proofs. With the proposed MUSTARD, we present a theorem-and-proof benchmark MUSTARDSAUCE with 5,866 valid data points. Each data point contains an informal statement, an informal proof, and a translated formal proof that passes the prover validation. We perform extensive analysis and demonstrate that MUSTARD generates validated high-quality step-by-step data. We further apply the MUSTARDSAUCE for fine-tuning smaller language models. The fine-tuned Llama 2-7B achieves a 15.41% average relative performance gain in automated theorem proving, and 8.18% in math word problems.[1]

## 1 INTRODUCTION

Large language models (LLMs) (OpenAI, 2023; 2022) have shown promising reasoning capabilities in various domains, including math word problem and theorem proving (Cobbe et al., 2021; Hendrycks et al., 2021; Zheng et al., 2022; Wu et al., 2021). These two tasks, which require strictly and successively multi-step inference, have become appeal domains to evaluate and develop LLMs' ability in complex reasoning. Recent works progress LLMs in solving math problems mainly through two techniques. The first is the chain-of-thoughts (CoT) prompting (Wei et al., 2022; Kojima et al., 2022; Wang et al., 2023b), which provides step-by-step solutions to the LLMs. The second is to leverage the LLMs' ability in code generation to generate formalized languages and utilize external solvers to obtain strict inference results (Wu et al., 2022; Jiang et al., 2023; Polu & Sutskever, 2020; Han et al., 2022; Polu et al., 2023). Both techniques rely on step-wise annotation to improve LLMs' performance and interpretability on the math problem.

Correct intermediate steps are crucial for LLMs to perform complex reasoning. However, high-quality step-wise annotations are hard to obtain, and Figure 1 demonstrates a few representative

---

[†]Corresponding author.
[1]Codes and data are available at: https://github.com/Eleanor-H/MUSTARD.

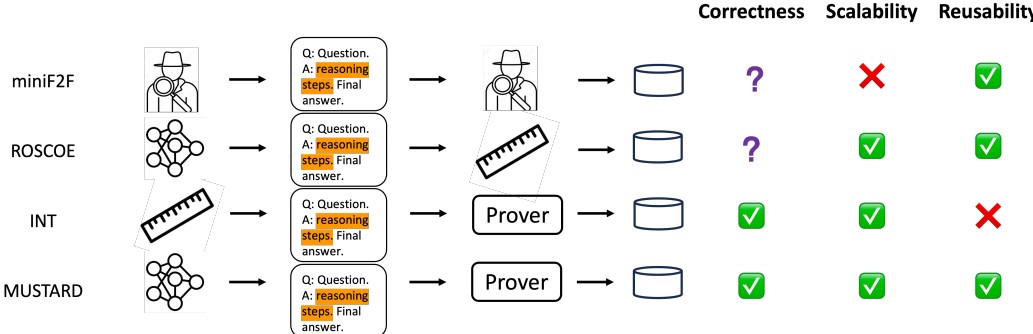

Figure 1: A comparison of methods of synthesizing and validating intermediate reasoning steps.

works. Previous works such as miniF2F (Zheng et al., 2022) resorts to manual annotation and validation to obtain high-quality step-wise labels. However, manual annotation requires heavy labor of knowledgeable experts, resulting in an extremely small-scale dataset. Manual checking also does not guarantee the correctness of data as the labelers would make mistakes in labeling. On the other hand, generating data with rule-based checking such as ROSCOE (Golovneva et al., 2023) can produce large-scale reasoning data. Given that the generated data are more friendly and readable for human beings, the correctness of the reasoning is not guaranteed by those rules. Moreover, another line of work such as INT (Wu et al., 2021) performs rule-based synthesis to generate validated proofs, which are both correct and large-scale. However, the data are brutally synthesized so that many generated proofs lack actual meaning. Therefore, we need a more efficient way to generate mathematical data that are large-scale, with accurate intermediate steps, and also meaningful mathematical knowledge to human beings.

To fill this gap, we propose MUSTARD, a data generation framework that uniformly synthesizes large-scale and high-quality mathematical data by combining the advantages of LLMs in verbalization and formal theorem provers in rigorous data validation. Specifically, MUSTARD first samples a few mathematical concepts from a predefined list and prompts an LLM to generate a related question described in natural language. Then, it applies the LLM to generate the corresponding solution in both natural and formal language. Given the generated solution, MUSTARD further validates them using a theorem prover. The passed one is considered to be correct and is a high-quality data point. The invalid one on the other hand is considered to be a challenging sample, which will be further combined with the error messages to prompt the LLM for a solution revision, and added as a challenging data point.

By applying the proposed MUSTARD one can obtain large amounts of problems and theorems with desired mathematical concepts and domains. Eventually, we build a mathematical dataset with validated informal and formal solutions, named MUSTARDSAUCE (MUSTARD resource).

We conduct extensive data analysis and experiments on the generated MUSTARDSAUCE. Through deep inspection of the data, we find that MUSTARD generates interesting and reasonable math problems by creatively combining two mathematical concepts, and MUSTARDSAUCE is diverse and has a high proportion of difficult data. We also observe that the prover is consistent with human evaluation, where humans usually consider a validated solution to have a higher quality than those without a formal validation process. Lastly, we fine-tune smaller-scale language models on MUSTARDSAUCE. The fine-tuned Llama 2-7B achieves improvements by 20.9% on zero-shot inference on GSM8K and achieves 8.7 of pass@1 on mathlib. These results demonstrate the effectiveness of MUSTARDSAUCE in improving the mathematical reasoning capabilities of language models.

The contributions of this paper are summarized as follows:

1. We propose a novel framework MUSTARD that can generate high-quality mathematical data (both informal and formal) with an interplay between generative language model and theorem prover assistants.
2. We release the MUSTARDSAUCE, which contains both math word problems and theorem-proving problems spanning over four educational levels. Each sample has corresponding informal and formal solutions.
3. We conduct extensive analysis and experiments on the generated data, demonstrating their quality, diversity, and effectiveness in improving language models' mathematical reasoning performance.

## 2   RELATED WORKS

**Large Language Models for Mathematical Reasoning**   The growing generative language models (Brown et al., 2020; OpenAI, 2022; 2023) show compelling potential for solving mathematical problems both in natural language proofs (OpenAI, 2023; Zheng et al., 2023a; Xiong et al., 2024) and in formal languages with theorem provers (Polu & Sutskever, 2020; Han et al., 2022; Zheng et al., 2023b; Polu et al., 2023). On the other hand, some works explore using language models to automatically translate natural language proofs into formal ones given few-shot demonstrations (Wu et al., 2022; Jiang et al., 2023; Liu et al., 2023; Wang et al., 2024). Chain-of-though reasoning (Wei et al., 2022; Kojima et al., 2022; Wang et al., 2023b) is demonstrated beneficial for the LLMs to derive correct answers. However, some recent works (Saparov & He, 2023; Golovneva et al., 2023) observe that the intermediate reasoning steps can be inconsistent. This paper proposes a data generation framework that taps the comprehensive mathematical reasoning capabilities of large language models. It generates mathematical reasoning problems with informal and formal solutions that are step-wise validated by a formal theorem prover. With the framework, we obtain high-quality mathematical data.

**Synthesizing Mathematical Data**   Obtaining large-scale high-quality mathematical data is a long-standing challenge. Previous data relies on well-trained annotators to hand-craft and review the formal proofs (Zheng et al., 2022), which is time and labour-consuming and results in a small data scale. Wang & Deng (2020) and Xiong et al. (2023) construct a neural generator for data synthesis, but it still requires the intervention of human-written data. Besides, Wu et al. (2021) explore using a theorem generator to automatically generate formal proofs with rules. However, the rule-based generation depends on given axioms in specified orders. As a result, the generated data is restricted to a few domains. On the other hand, recent works demonstrate the effectiveness of distilling knowledge from large language models (West et al., 2022; Yuan et al., 2023; Li et al., 2023), and some of them (Wang et al., 2023c; Xu et al., 2023) explore data evolution by properly prompting the language models. The proposed framework explores eliciting mathematical knowledge from large language models to achieve diverse and large-scale mathematical data. In this framework, an interplay between the language model and a formal proof assistant controls the quality and difficulties of data. Using the proposed framework, we collect a large-scale mathematical dataset that contains diverse and multiple-difficulty math questions with high-quality solutions.

## 3   MUSTARD

In this work, we aim to obtain large-scale mathematical data with multi-step annotations and propose MUSTARD to generate diverse and high-quality math and theorem-proving problems with multi-step informal and formal solutions. As shown in Figure 2, MUSTARD consists of three stages. In the first concept seeding stage, MUSTARD samples a set of math concepts as the problem domain. Then in the second solution generation stage, it generates the concept-related problem and solution by prompting an LLM. In the third stage, a theorem prover is used to validate the generated solution. If the solution can not pass the prover, the error message is returned to the second stage for another turn of solution generation. Through interaction between the LLM and a formal proof assistant, MUSTARD can generate diverse and high-quality data that contains both informal and formal solutions. We describe the details of each stage in this section.

### 3.1   CONCEPT SEEDING

We first define and build a mathematical concept pool that covers as complete sub-subjects in mathematics and educational levels as possible. Specifically, we collect all math courses on the Khan Academy website[2], the large-scale online educational platform. The resulting pool includes concepts in four educational levels: elementary school, middle school, high school, and higher education. Each educational level has 5 to 9 math domains, covering different types of math problems such as algebra and geometry. Each domain contains subdivided mathematical concepts to inspect different mathematical abilities like polynomial arithmetic or factorization. Concept statistics and detailed concepts in each domain are demonstrated in Appendix B.

Given the concept pool, for each educational level, MUSTARD uniformly samples 1 or 2 concepts from all domains as seeds, and then generates mathematical problems that cover the concepts. In particular, given an educational level, taking 2 concepts from different subjects challenges the model to generate problems that join diverse domains while keeping the problems reasonable.

---

[2]https://www.khanacademy.org/math

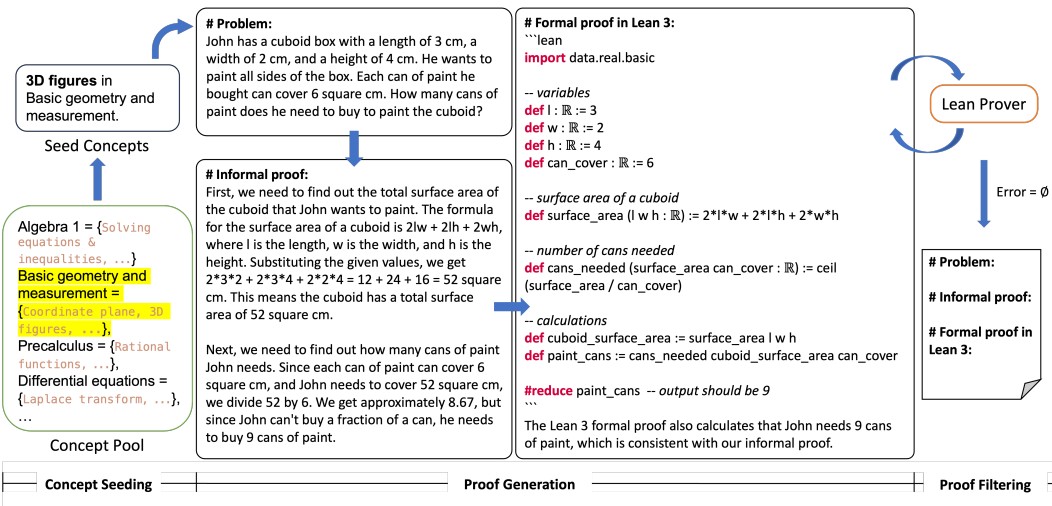

Figure 2: Overview of MUSTARD. The mathematical theorem (indicated by "# Problem:") is created according to the randomly selected seed concepts. Following the theorem, a corresponding informal proof is generated, which is then translated into a formal proof in Lean 3. The formal proof is passed into Lean Prover for quality feedback, and according to the error messages the formal proof is revised and improved. The proofs that pass the Lean Prover are regarded as high-quality data points and collected.

## 3.2 PROOF GENERATION

Given the sampled mathematical concepts, MUSTARD generates math problems and their corresponding solutions. Specifically, MUSTARD leverages the capability of LLMs in generating natural language and code and prompts an LLM to generate the problem statement, its natural language solution, and a formal solution written in Lean. As a result, the LLM needs to complete the following three tasks: (T1) Generating a math problem that relates to the given concepts; (T2) Solving the math problem with a natural language proof; (T3) Performing auto-formalization to translate the written natural language proof into a formalized proof. In this work, we use GPT-4 OpenAI (2023) as the LLM for proof generation.

We intend to generate a problem based on educational level, math domains, and concepts. Considering that mathematical problems include proof and calculation, we also introduce the question types into the prompt for generating theorem proving and word problems respectively. Moreover, we do not include any exemplars or other manual interventions except for the sampled concepts. We intend to avoid potential biases brought by the concepts inside the exemplars and achieve a more diverse generation. The prompt template is shown as the following:

```
You are a math expert. Now please come up with a math problem according to the following
requirements. The math problem should contain a question part (indicated by ``Problem:
''), a corresponding solution in natural language (indicated by ``Informal proof:''), and a
translated formal solution in Lean 3 (indicated by ``Formal proof in Lean 3:''). Please note
that the informal proof and the formal proof need to be identical.

Please create a [QUESTION TYPE] in the level of [EDUCATIONAL LEVEL] based on the following
knowledge point(s): [CONCEPT] in [DOMAIN]; [CONCEPT] in [DOMAIN].

You must respond in the following format:

# Problem: ...

# Informal proof: ...

# Formal proof in Lean 3: ...
```

The "[]" indicates the placeholders for the corresponding question type, educational level, concepts, and domains. Multiple concepts are separated by ";". We retrieved the text after the "Problem:", "Informal proof:" and "Formal proof in Lean 3:" as the generated sample.

### 3.3 Proof Filtering

In the proof-filtering stage, MUSTARD interacts with the Lean Prover (de Moura et al., 2015) to obtain validation messages of the proof steps, which guides data revision and filtering. Specifically, after a formal solution is passed to the Lean Prover and if the prover returns no error message, the corresponding data point is collected into the valid dataset. Otherwise, MUSTARD collects the error messages from the prover and prompts the language model to revise the invalid solution. To help the language model locate the incorrect lines described in the error messages, we also add a line number at the beginning of each line in the formal solutions. The verification and self-refinement are performed in multiple rounds until LLM generates a valid solution. We use the number of rounds to measure the difficulty of the generated sample, assuming a difficult problem is hard to solve by an LLM and requires more rounds of correction. The prompt template of a single round of correction is demonstrated as follows, and the complete prompt template is shown in Table 13 in Appendix C.1:

```
# Formal proof (c) in Lean 3:
```lean
line 1 
line 2 
line 3 
...
```

# Error messages for Formal proof (c) from Lean Prover:
<error messages>
```

## 4 Experiments

### 4.1 Case Study

We first inspect the data points generated by MUSTARD. Table 1 shows a generated math problem in which MUSTARD creatively combines two mathematical concepts and constructs a reasonable question. The generated question includes knowledge from both concepts. It is suggested that MUSTARD can join the concepts and construct a reasonable question. Furthermore, Table 2 demonstrates a case that MUSTARD provides solid and comprehensive solutions in both natural language and Lean. Although the formal proof is long, it is consistent and passes the prover's validation. It is demonstrated that MUSTARD can generate long valid solutions.

### 4.2 Human Evaluation

To further explore the quality of the data generated by MUSTARD, we recruit professionals who have expertise in mathematics and the Lean language to perform a sanity check on the data points. We randomly select 200 data points from the generated data, 100 of which pass the Lean Prover (Group Valid) and 100 of which do not (Group Invalid). The sanity check covers the four sections in each data point (i.e., informal statement, informal proof, formal statement, and formal proof), and includes factuality check and consistency check. Specifically, a high-quality data point should have a factually correct informal statement (D1) and a correct solution (D4). The formal statement and proof should be aligned with the informal descriptions (D5, D6). Moreover, the desired data point should meet the specified seed concepts (D2) and question type (D3). The six inspection dimensions and their requirements are demonstrated in Table 3. A data point is scored 1 in a dimension if it meets the requirement, otherwise, it gets 0.

The accuracies of Group Valid and Group Invalid in each dimension are demonstrated in Table 3. We also report the corresponding $p$-value in each dimension. (D4) and (D6) show significant differences in accuracy between the two groups. The results indicate that high-quality data points have significantly better auto-formalization results. As a result, given the validation of formal proofs by the Lean Prover, the data points have guaranteed high-quality informal proofs. Moreover, (D1) also shows significance with the inspected data scaled up. The differences in statement alignment (D5) and informal statement relevance (D2) of the two groups are less significant. Furthermore, no significant differences are observed in question type classification (D3), which indicates that Lean Prover validation in MUSTARD does not significantly influence the classification. Overall, the human evaluation results suggest that formally validated data have significantly higher quality.

### 4.3 Data Quality by Downstream Application

To evaluate the impact of MUSTARDSAUCE on enhancing mathematical reasoning abilities, we use the data to fine-tune smaller-scale language models and evaluate them on math word problems (MWP) and automated theorem proving (ATP). Specifically, given all the generated data, we first

Table 1: An informal statement generated by MUSTARD. The complete data point is demonstrated in Table 18 in Appendix E.

| **Question Type: Theorem Proving.   Educational Level: Middle School.   k=2.** |
| --- |
| **Concept(s):** *Geometry* in 8th grade; *Algebraic expressions* in Algebra basics. |
| **Informal Statement**
Given a rectangle ABCD where AB is x + 5 and AD is 2x - 3. Prove that the area of the rectangle ABCD is $(2x^2 + 7x - 15)$ square units. |

Table 2: A data point generated by MUSTARD.

| **Question Type: Word Problem.   Educational Level: Higher Education.   k=1.** |
| --- |
| **Concept(s):** *Series* in Integral Calculus. |

**Informal Statement**

Prove that the integral of the series $\sum$ (from $n = 1$ to infinity) of $\frac{-1^{(n+1)}}{n}$ equals $\ln(2)$.

**Informal Proof**

We start by recognizing the series $\sum$ (from $n = 1$ to infinity) of $\frac{-1^{(n+1)}}{n}$ as the alternating harmonic series, which is known to converge to $\ln(2)$.

**Formal Statement and Proof**
```
import data.real.basic
import analysis.special_functions.pow

open real

lemma integral_of_series :
 ∫ x in 0..1, (ln (1 + x)) = ∑' (n :  ℕ), ((-1)^(n+1) / (n + 1)) :=
begin
 have h_der :  ∀ x ∈ set.Icc (0 :  ℝ) 1, deriv (λ x, (ln (1 + x))) x = 1
/ (1 + x),
 { intros x hx,
 have h_pos :  0 < 1 + x := by nlinarith,
 exact deriv_ln_one_add h_pos },

 have h_int :  interval_integral (λ x, 1 / (1 + x)) 0 1 volume = ∑' (n :
ℕ), ((-1)^(n+1) / (n + 1)),
 { have h_frac :  ∀ (n :  ℕ), ∫ x in 0..1, x^n = 1 / (n + 1),
 { intro n,
 calc ∫ x in 0..1, x^n = [x^(n+1) / (n+1)] | 0..1 :  integral_pow n
 ...  = 1 / (n + 1) :  by { rw integral_interval, simp } },
 rw [interval_integral.integral_of_le, h_frac],
 simp,
 linarith },

 have h_eq :  ∫ x in 0..1, (ln (1 + x)) = ∫ x in 0..1, 1 / (1 + x),
 { congr, ext, exact h_der x (set.mem_Icc.mpr ⟨by nlinarith, by
nlinarith⟩) },

 rw [h_eq, h_int],
end
```

have 5,866 valid data points that pass the Lean Prover. We denote this subset MUSTARDSAUCE-valid. We then extract the same number of invalid data points as the subset of MUSTARDSAUCE-invalid, and extract an equal size of random subset MUSTARDSAUCE-random. Each MUSTARDSAUCE subset contains 5,866 data points. Moreover, we randomly split MUSTARDSAUCE-valid into 4,866 training data, 500 validation data, and 500 test data for benchmarking model performances on the dataset. We denote the test set as MUSTARDSAUCE-test. Furthermore, we also test on the entire generated data set with 28,316 data points, which we denote MUSTARDSAUCE-tt.

We employ LoRA (Hu et al., 2021) for fine-tuning the open-source GPT2-large (Radford et al., 2019), Llama 2-7B and Llama 2-70B (Touvron et al., 2023) on each MUSTARDSAUCE subset. De-

Table 3: Inspection dimensions and requirements in human evaluation. IS: Informal Statement. IP: Informal Proof. FS: Formal Statement. FP: Formal Proof. RT: Reasoning Type. Significant $p < 0.005$ are marked with **bold**.

| Inspection Dimension | Requirement | Valid | Invalid | $p$-value |
|---|---|---|---|---|
| (D1) IS Correctness | *Whether the informal statement is factually correct.* | 93.50 | 83.50 | **0.00167** |
| (D2) IS Relevance | *Whether the informal statement is relevant to each seed concept.* | 87.50 | 92.50 | 0.09604 |
| (D3) RT Classification | *Whether the informal statement is of the required question type.* | 67.00 | 68.50 | 0.74903 |
| (D4) IP Correctness | *Whether the informal proof correctly solves the informal statement.* | 88.50 | 73.50 | **0.00012** |
| (D5) IS-FS Alignment | *Whether the informal statement and the formal statement describe the same problem and are aligned with each other.* | 74.00 | 66.50 | 0.10138 |
| (D6) IP-FP Alignment | *Whether the informal proof and the formal proof describe the same solution and have aligned proof steps.* | 72.00 | 54.00 | **0.00018** |

Table 4: Maj1@1 results on GSM8K (G) and MATH (M). Zero: Zero-shot. Few: Few-shot. $>$ denotes a fine-tuning step. `in`: MUSTARDSAUCE-invalid. `ra`: MUSTARDSAUCE-random. `va`: MUSTARDSAUCE-valid. `tt`: MUSTARDSAUCE-tt. `gt`: GSM8K training split.

| MODEL | Zero (G) | Few (G) | Zero (M) | Few (M) | MODEL | Zero (G) | Few (G) | Zero (M) | Few (M) |
|---|---|---|---|---|---|---|---|---|---|
| *Baselines* | | | | | | | | | |
| GPT2-large | 3.4 | 5.1 | 0.6 | 1.0 | GPT2-large $>$ `gt` | 14.6 | 17.4 | 4.6 | 6.8 |
| Llama 2-7B | 7.2 | 12.8 | 2.0 | 2.6 | Llama 2-7B $>$ `gt` | 24.5 | 28.2 | 10.4 | 12.6 |
| *Fine-tuning* | | | | | | | | | |
| GPT2-large $>$ `tt` | 4.2 | 6.8 | 1.4 | 2.4 | GPT2-large $>$ `tt` $>$ `gt` | 16.1 | 18.9 | 4.8 | 8.0 |
| GPT2-large $>$ `in` | 3.9 | 6.4 | 1.2 | 2.0 | GPT2-large $>$ `in` $>$ `gt` | 15.4 | 17.7 | 4.4 | 7.6 |
| GPT2-large $>$ `ra` | 4.1 | 6.7 | 1.4 | 2.2 | GPT2-large $>$ `ra` $>$ `gt` | 15.7 | 18.5 | 4.8 | 7.8 |
| GPT2-large $>$ `va` | 4.6 (+12.20%) | 7.0 (+4.48%) | 1.8 (+28.57%) | 2.8 (+27.27%) | GPT2-large $>$ `va` $>$ `gt` | 16.5 (+5.10%) | 20.1 (+8.65%) | 5.6 (+16.67%) | 8.4 (+7.69%) |
| Llama 2-7B $>$ `tt` | 9.6 | 16.0 | 3.2 | 3.8 | Llama 2-7B $>$ `tt` $>$ `gt` | 27.4 | 31.5 | 13.0 | 14.4 |
| Llama 2-7B $>$ `in` | 9.1 | 14.9 | 2.4 | 3.2 | Llama 2-7B $>$ `in` $>$ `gt` | 26.9 | 30.3 | 12.4 | 13.8 |
| Llama 2-7B $>$ `ra` | 9.5 | 15.4 | 3.0 | 3.6 | Llama 2-7B $>$ `ra` $>$ `gt` | 27.1 | 30.7 | 12.6 | 14.2 |
| Llama 2-7B $>$ `va` | 10.3 (+8.42%) | 16.9 (+9.74%) | 3.2 (+6.67%) | 4.2 (+16.67%) | Llama 2-7B $>$ `va` $>$ `gt` | 27.9 (+2.95%) | 32.5 (+5.86%) | 13.8 (+9.52%) | 15.0 (+5.63%) |

tailed model configuration and training procedure are described in Appendix F. For the task of math word problems, we use GSM8K (Cobbe et al., 2021) and MATH dataset Hendrycks et al. (2021)[3] for evaluation. For evaluating automated theorem proving, we use Mathlib[4] and the miniF2F (Zheng et al., 2022) benchmark. We also evaluate models on MUSTARDSAUCE-test after being fine-tuned on the MUSTARDSAUCE-valid training split. Tables 4 and 5 demonstrate the model performances. We also follow Han et al. (2022) to ablate the fine-tuning steps and demonstrate the results in Table 6.

In general, fine-tuning the models on MUSTARDSAUCE improves the mathematical reasoning of the models. On average, we have an 18.15% relative performance gain after fine-tuning with MUSTARDSAUCE-valid compared with fine-tuning with MUSTARDSAUCE-random in ATP (Table 5) and 11.01% in MWP (Table 4). The fine-tuned Llama 2-7B achieves average gains of 15.41% and 8.18% on ATP and MWP, and the fine-tuned GPT 2-large 20.89% and 15.41%, respectively. Specifically, in ATP, the Llama 2-7B achieves significant performance gains of 16.00% on both mathlib and miniF2F, while increasing 17.31% on the MUSTARDSAUCE-test. In MWP, the performance improvements are also consistent in two datasets and both zero-shot and few-shot inference. We further compare the results fine-tuned with MUSTARDSAUCE-tt and MUSTARDSAUCE-valid. We find that models fine-tuned with the entire generated data are inferior to models fine-tuned with MUSTARDSAUCE-valid. Although the increase in the amount of fine-tuned data makes the model perform better compared to fine-tuning on MUSTARDSAUCE-invalid and MUSTARDSAUCE-

---

[3]Specifically, we use the released test split with 500 problems in PRM800K (Lightman et al., 2023) which is demonstrated to be representative of the MATH test set as a whole.

[4]https://github.com/leanprover-community/mathlib

Table 5: Pass@1 results on automated theorem proving tasks. > denotes a fine-tuning step. `test`: MUSTARDSAUCE-test. `in`: MUSTARDSAUCE-invalid. `ra`: MUSTARDSAUCE-random. `va`: MUSTARDSAUCE-valid. `tt`: MUSTARDSAUCE-tt. `mt`: mathlib training split. Note that the reported results on MUSTARDSAUCE-test are obtained by only fine-tuning on the MUSTARD-SAUCE-valid training split.

| MODEL | mathlib | miniF2F | test | MODEL | mathlib | miniF2F | test |
|---|---|---|---|---|---|---|---|
| *Baselines* | | | | | | | |
| GPT2-large | 0.0 | 0.0 | 0.0 | GPT2-large > mt | 5.6 | 2.9 | 8.6 |
| Llama 2-7B | 0.0 | 0.0 | 0.0 | Llama 2-7B > mt | 14.3 | 7.0 | 10.8 |
| *Fine-tuning* | | | | | | | |
| GPT2-large > in | 2.0 | 0.0 | 6.0 | GPT2-large > in> mt | 5.9 | 2.0 | 8.2 |
| GPT2-large > ra | 3.0 | 1.2 | 7.0 | GPT2-large > ra> mt | 6.6 | 2.9 | 9.6 |
| GPT2-large > va | 3.7 | 1.6 | 8.3 | GPT2-large > va> mt | 7.4 | 3.7 | 10.6 |
| | (+23.33%) | (+33.33%) | (+18.57%) | | (+12.12%) | (+27.59%) | (+10.42%) |
| Llama 2-7B > tt | 8.3 | 2.6 | 11.7 | Llama 2-7B > tt> mt | 15.1 | 7.0 | 13.6 |
| Llama 2-7B > in | 5.8 | 1.2 | 8.6 | Llama 2-7B > in> mt | 11.6 | 5.7 | 12.6 |
| Llama 2-7B > ra | 7.5 | 2.5 | 10.4 | Llama 2-7B > ra> mt | 14.7 | 6.6 | 13.2 |
| Llama 2-7B > va | 8.7 | 2.9 | 12.2 | Llama 2-7B > va> mt | 15.7 | 7.8 | 14.4 |
| | (+16.00%) | (+16.00%) | (+17.31%) | | (+6.80%) | (+18.18%) | (+18.18%) |

Table 6: Ablation study of different fine-tuning settings. `test`: MUSTARDSAUCE-test. `va`: MUSTARDSAUCE-valid. `mt`: mathlib training split.

| MODEL | test |
|---|---|
| GPT2-large > va | 8.3 |
| GPT2-large > va > mt | 10.6 |
| GPT2-large > mt > va | 9.8 |
| Llama 2-7B > va | 12.2 |
| Llama 2-7B > va > mt | 14.4 |
| Llama 2-7B > mt > va | 13.8 |

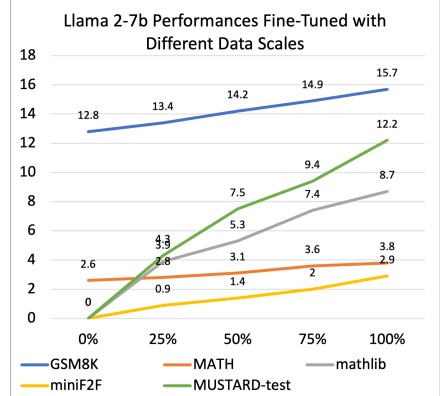

Figure 3: Model performances fine-tuned with different data scales.

random, the model's performance still lags behind that of fine-tuning on smaller amounts but higher quality data. Therefore, our proposed framework that introduces the theorem prover is effective and beneficial. Furthermore, complementary experimental results of larger Llama 2-70B are demonstrated in Table 28 in Appendix G. The results suggest that our method remains effective when fine-tuning a larger language model.

### 4.4 IMPACT OF DATA SCALABILITY

To further study the impact of data scale on the fine-tuning results, we randomly sample 75%, 50%, 25%, and 0% data from MUSTARDSAUCE-valid and fine-tune Llama 2-7B. The results are shown in Figure 3. In general, the results on all datasets increase as the fine-tuning data scales up. Specifically, performances on the MUSTARD-test and mathlib have the most significant growth without a decrease in the growth rate. Therefore we expect further performance improvements when more high-quality data are included.

### 4.5 PASS RATE

We study the mathematical generation ability of MUSTARD by investigating its pass rates on generating valid data points. The pass@1 results of the generated formal proofs of GPT-4 OpenAI (2023) and GPT-3.5 OpenAI (2022) are shown in Table 7. We have the following observations. First of all, the overall pass@1 results are high, showing the LLMs especially GPT-4 capable of performing zero-shot mathematical reasoning. Second, the pass rates of word problems are generally higher than those of theorem proving. It indicates that the word problems are relatively easier and more familiar to the LLMs, while theorem proving is more challenging. Third, the all-at-once generation and step-by-step generation have similar pass rates at lower educational levels. For more challenging questions such as those at the high school level and higher educational level, step-by-step generation shows slight advantages over the all-at-once generation. This indicates that dividing and conquering (T1), (T2), and (T3) helps the model to generate higher-quality formal proofs, but the improvement is limited. Last but not least, the improvements in pass rates after 1-step and 2-

Table 7: Pass@1 of formal proofs generated by MUSTARD. k: Number of seed concepts.

| | | | Thoerem Proving | | | | | Word Problem | | | |
|---|---|---|---|---|---|---|---|---|---|---|---|
| | | | All (GPT-4) | | Step (GPT-4) | All (GPT-3.5) | | All (GPT-4) | | Step (GPT-4) | All (GPT-3.5) |
| | | #correct=0 | 1 (Δ) | 2 (Δ) | 0 | 0 | 0 | 1 (Δ) | 2 (Δ) | 0 | 0 |
| k=1 | elem | 26.0 | 48.0 (+22.0) | 55.9 (+33.9) | 25.5 | 15.1 | 38.0 | 59.7 (+21.7) | 67.0 (+45.3) | 40.1 | 22.2 |
| | midd | 16.4 | 31.8 (+15.4) | 39.6 (+24.2) | 17.0 | 4.3 | 22.9 | 39.7 (+16.8) | 47.4 (+30.6) | 28.4 | 7.0 |
| | high | 6.8 | 14.4 (+7.6) | 17.2 (+9.6) | 6.6 | 1.9 | 6.7 | 16.8 (+10.1) | 21.9 (+11.8) | 8.1 | 3.4 |
| | higher | 2.1 | 5.6 (+3.5) | 9.8 (+6.3) | 3.0 | 0.7 | 3.6 | 10.9 (+7.3) | 16.0 (+8.7) | 4.3 | 2.7 |
| k=2 | elem | 24.1 | 42.3 (+18.2) | 52.3 (+34.1) | 23.2 | 12.8 | 32.2 | 49.5 (+17.3) | 58.2 (+40.9) | 31.9 | 22.1 |
| | midd | 14.0 | 25.2 (+11.2) | 34.1 (+22.9) | 15.0 | 5.3 | 16.9 | 27.3 (+10.4) | 34.5 (+24.1) | 17.3 | 5.7 |
| | high | 3.8 | 8.3 (+4.5) | 12.1 (+7.6) | 5.4 | 2.2 | 5.7 | 11.0 (+5.3) | 16.2 (+10.9) | 4.5 | 2.5 |
| | higher | 1.1 | 3.3 (+2.2) | 5.0 (+2.8) | 2.6 | 1.3 | 2.6 | 7.0 (+4.4) | 10.5 (+6.1) | 2.9 | 2.1 |

Figure 4: Distributions of formal proof lengths.

step corrections are significant. For example, theorem proving at elementary-school level with 1 seed concept improves by 22.0% after 1-step correction, and 33.9% after 2-step correction. Word problem at the elementary-school level with 1 seed concept improves 45.3% after 2-step correction. The most difficult setting of generating theorem proving data at the higher-educational level with 2 seed concepts achieves 2.8% improvement after 2-step correction, and the word problem counterpart achieves 6.1% improvement. This indicates that the LLMs have a great potential for self-correction given the error message feedback and limited instructions.

## 4.6 DIVERSITY AND DIFFICULTY

We compute ROUGE-L (Lin, 2004) to check the diversity of generated informal statements and proofs. The resulting ROUGE-L scores are below 0.25 and indicate high data diversity. We demonstrate detailed computation and results in Appendix D.2. We then investigate the proof lengths in MUSTARDSAUCE and the distributions are demonstrated in Figure 4. We count both reasoning steps of formal statement-proof pairs and steps of formal proof only, which are shown on the left- and right-hand side of Figure 4, respectively. It is demonstrated that proof length increases over educational levels. Solving elementary problems needs about 5 to 10 steps while solving higher-educational problems requires a median number between 10 to 15 steps. The most challenging problems require around 30 reasoning steps or about 20 formal proof steps. Therefore, MUSTARD-SAUCE produces diverse mathematical problems with multiple topics and difficulties.

## 5 CONCLUSION

In this paper, we introduce MUSTARD to automatically generate mathematical datasets with high-quality solutions that cover a variety of mathematical skills. Leveraging the LLM and Lean Prover, MUSTARD can generate the problem statement, informal solution, and formal solution. and use a Lean Prover to automatically verify the formal solution and provide feedback for revision. At last, we apply the proposed MUSTARD and obtain 5,866 problems with step-by-step solutions that cover different educational levels and mathematical abilities. The obtained dataset has shown its high quality, diversity, and effectiveness in improving language models' mathematical reasoning performance, showing the great potential of our proposed MUSTARD and dataset in further research on language models.

## ACKNOWLEDGEMENTS

This work was supported in part by the National Key R&D Program of China under Grant No. 2020AAA0109700, Guangdong Outstanding Youth Fund (Grant No. 2021B1515020061), National Natural Science Foundation of China (NSFC) under Grant No.61976233, Mobility Grant Award under Grant No. M-0461, Shenzhen Science and Technology Program (Grant No. GJHZ20220913142600001), Nansha Key RD Program under Grant No.2022ZD014, National Natural Science Foundation of China under Grant No.62006255, and the National Natural Science Foundation of China under Grant 62371411, the Research Grants Council of the Hong Kong SAR under Grant GRF 11217823, InnoHK initiative, the Government of the HKSAR, Laboratory for AI-Powered Financial Technologies. We thank MindSpore for the partial support of this work, which is a new deep learning computing framework[5].

The authors would also like to thank Hui Jin, Jianhao Shen, Chengwu Liu, Cen Li, and Junhao Cheng for their hard work on the manual check related to Section 4.2.

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

## A  FUTURE WORKS

Current formal validation still suffers mild inconsistency between the #reduce statements and various kinds of theorem proofs. In the future, we will explore more rigorous and careful data filtering. We will also explore data generation and mathematical reasoning via the same language model, which is an interesting setup to study large language models' proficiency of mathematical reasoning. Moreover, the ablation study on data scalability shows consistent performance increases when more data from MUSTARD are introduced, suggesting a great potential for scalability. Fortunately, MUSTARD reduces the cost of acquiring such high-quality step-by-step complex reasoning data and obtains correct, scalable, and reusable data. Therefore, in future work, we would love to build a community in which all members can join the data synthesis process, and acquire and share more high-quality data with the whole community.

## B  MATHEMATICAL CONCEPTS

Table 8 shows the concept statistics. Concepts are grouped into multiple domains mainly according to subjects, except that at the elementary level are grouped by grades due to the lack of subject division. Tables 9, 10, 11 and 12 demonstrate the detailed concepts in each domain.

Table 8: Statistics of the mathematical concept pool.

| Elementary School | | Middle School | | High School | | Higher Education | |
|---|---|---|---|---|---|---|---|
| Domains | # | Domains | # | Domains | # | Domains | # |
| 1st grade | 3 | 7th grade | 7 | Algebra 1 | 16 | AP College Statistics | 14 |
| 2nd grade | 8 | 8th grade | 7 | Algebra 2 | 12 | College Algebra | 14 |
| 3rd grade | 14 | Algebra basics | 8 | High school geometry | 9 | Differential Calculus | 6 |
| 4th grade | 14 | Pre-algebra | 15 | Trigonometry | 4 | Integral Calculus | 5 |
| 5th grade | 16 | Basic geometry and measurement | 14 | Statistics and probability | 16 | AP College Calculus AB | 10 |
| 6th grade | 8 | | | High school statistics | 7 | AP College Calculus BC | 12 |
| | | | | Precalculus | 10 | Multivariable calculus | 5 |
| | | | | Calculus 1 | 8 | Differential equations | 3 |
| | | | | Calculus 2 | 6 | Linear algebra | 3 |
| # Domain | 6 | | 5 | | 9 | | 9 |
| # Concept | 63 | | 51 | | 88 | | 72 |

Table 9: Mathematical concepts at the level of elementary school.

| Domain | Concept |
|---|---|
| 1st grade | "Place value", "Addition and subtraction", "Measurement, data, and geometry", |
| 2nd grade | "Add and subtract within 20", "Place value", "Add and subtract within 100", "Add and subtract within 1,000", "Money and time", "Measurement", "Data", "Geometry", |
| 3rd grade | "Intro to multiplication", "1-digit multiplication", "Addition, subtraction, and estimation", "Intro to division", "Understand fractions", "Equivalent fractions and comparing fractions", "More with multiplication and division", "Arithmetic patterns and problem solving", "Quadrilaterals", "Area", "Perimeter", "Time", "Measurement", "Represent and interpret data", |
| 4th grade | "Place value", "Addition, subtraction, and estimation", "Multiply by 1-digit numbers", "Multiply by 2-digit numbers", "Division", "Factors, multiples and patterns", "Equivalent fractions and comparing fractions", "Add and subtract fractions", "Multiply fractions", "Understand decimals", "Plane figures", "Measuring angles", "Area and perimeter", "Units of measurement", |
| 5th grade | "Decimal place value", "Add decimals", "Subtract decimals", "Add and subtract fractions", "Multi-digit multiplication and division", "Multiply fractions", "Divide fractions", "Multiply decimals", "Divide decimals", "Powers of ten", "Volume", "Coordinate plane", "Algebraic thinking", "Converting units of measure", "Line plots", "Properties of shapes", |
| 6th grade | "Ratios", "Arithmetic with rational numbers", "Rates and percentages", "Exponents and order of operations", "Negative numbers", "Variables & expressions", "Equations & inequalities", "Plane figures", |

Table 10: Mathematical concepts at the level of middle school.

| Domain | Concept |
| --- | --- |
| 7th grade | "Negative numbers: addition and subtraction", "Negative numbers: multiplication and division", "Fractions, decimals, & percentages", "Rates & proportional relationships", "Expressions, equations, & inequalities", "Geometry", "Statistics and probability", |
| 8th grade | "Numbers and operations", "Solving equations with one unknown", "Linear equations and functions", "Systems of equations", "Geometry", "Geometric transformations", "Data and modeling", |
| Algebra basics | "Foundations", "Algebraic expressions", "Linear equations and inequalities", "Graphing lines and slope", "Systems of equations", "Expressions with exponents", "Quadratics and polynomials", "Equations and geometry", |
| Pre-algebra | "Factors and multiples", "Patterns", "Ratios and rates", "Percentages", "Exponents intro and order of operations", "Variables & expressions", "Equations & inequalities introduction", "Percent & rational number word problems", "Proportional relationships", "One-step and two-step equations & inequalities", "Roots, exponents, & scientific notation", "Multi-step equations", "Two-variable equations", "Functions and linear models", "Systems of equations", |
| Basic geometry and measurement | "Intro to area and perimeter", "Intro to mass and volume", "Measuring angles", "Plane figures", "Units of measurement", "Volume", "Coordinate plane", "Decomposing to find area", "3D figures", "Circles, cylinders, cones, and spheres", "Angle relationships", "Scale", "Triangle side lengths", "Geometric transformations", |

## C  PROMPT TEMPLATES IN MUSTARD

### C.1  PROMPT TEMPLATE FOR PROOF FILTERING

Table 13 demonstrates the prompt template used in the proof-filtering stage in MUSTARD.

### C.2  PROMPT TEMPLATES FOR STEP-BY-STEP GENERATION

Table 14 demonstrates the variation of prompt templates used in the proof-generation stage in MUSTARD. In this variation, an LLM is conducted to perform (T1), (T2), and (T3) separately to generate the informal statement, informal solution, and formal solution. It is noted that to prompt the LLM to fulfill (T3), we assign the character of "a master in Lean" rather than the previous "a math expert" to obtain higher quality Lean proofs.

Table 11: Mathematical concepts at the level of high school.

| Domain | Concept |
|---|---|
| Algebra 1 | "Algebra foundations", "Solving equations & inequalities", "Working with units", "Linear equations & graphs", "Forms of linear equations", "Systems of equations", "Inequalities (systems & graphs)", "Functions", "Sequences", "Absolute value & piecewise functions", "Exponents & radicals", "Exponential growth & decay", "Quadratics: Multiplying & factoring", "Quadratic functions & equations", "Irrational numbers", "Creativity in algebra", |
| Algebra 2 | "Polynomial arithmetic", "Complex numbers", "Polynomial factorization", "Polynomial division", "Polynomial graphs", "Rational exponents and radicals", "Exponential models", "Logarithms", "Transformations of functions", "Equations", "Trigonometry", "Modeling", |
| High school geometry | "Performing transformations", "Transformation properties and proofs", "Congruence", "Similarity", "Right triangles & trigonometry", "Analytic geometry", "Conic sections", "Circles", "Solid geometry", |
| Trigonometry | "Right triangles & trigonometry", "Trigonometric functions", "Non-right triangles & trigonometry", "Trigonometric equations and identities", |
| Statistics and probability | "Analyzing categorical data", "Displaying and comparing quantitative data", "Summarizing quantitative data", "Modeling data distributions", "Exploring bivariate numerical data", "Study design", "Probability", "Counting, permutations, and combinations", "Random variables", "Sampling distributions", "Confidence intervals", "Significance tests (hypothesis testing)", "Two-sample inference for the difference between groups", "Inference for categorical data (chi-square tests)", "Advanced regression (inference and transforming)", "Analysis of variance (ANOVA)", |
| High school statistics | "Displaying a single quantitative variable", "Analyzing a single quantitative variable", "Two-way tables", "Scatterplots", "Study design", "Probability", "Probability distributions & expected value", |
| Precalculus | "Composite and inverse functions", "Trigonometry", "Complex numbers", "Rational functions", "Conic sections", "Vectors", "Matrices", "Probability and combinatorics", "Series", "Limits and continuity", |
| Calculus 1 | "Limits and continuity", "Derivatives: definition and basic rules", "Derivatives: chain rule and other advanced topics", "Applications of derivatives", "Analyzing functions", "Integrals", "Differential equations", "Applications of integrals", |
| Calculus 2 | "Integrals review", "Integration techniques", "Differential equations", "Applications of integrals", "Parametric equations, polar coordinates, and vector-valued functions", "Series", |

# D  MORE STATISTIC RESULTS OF MUSTARDSAUCE

## D.1  DIFFICULTY OF MUSTARDSAUCE BY NUMBER OF CORRECTION

Figure 5 demonstrates the proportions of data points that obtain valid proof after different numbers of corrections. Generally speaking, data points without correction are relatively less difficult for the LLMs, while those that require multiple corrections are challenging. Overall, theorem-proving problems are more challenging for LLMs to solve than the generated word problems. For data

Table 12: Mathematical concepts at the level of higher education.

| Domain | Concept |
| --- | --- |
| AP College Statistics | "Exploring categorical data", "Exploring one-variable quantitative data: Displaying and describing", "Exploring one-variable quantitative data: Summary statistics", "Exploring one-variable quantitative data: Percentiles, z-scores, and the normal distribution", "Exploring two-variable quantitative data", "Collecting data", "Probability", "Random variables and probability distributions", "Sampling distributions", "Inference for categorical data: Proportions", "Inference for quantitative data: Means", "Inference for categorical data: Chi-square", "Inference for quantitative data: slopes", "Prepare for the 2022 AP Statistics Exam", |
| College Algebra | "Linear equations and inequalities", "Graphs and forms of linear equations", "Functions", "Quadratics: Multiplying and factoring", "Quadratic functions and equations", "Complex numbers", "Exponents and radicals", "Rational expressions and equations", "Relating algebra and geometry", "Polynomial arithmetic", "Advanced function types", "Transformations of functions", "Rational exponents and radicals", "Logarithms", |
| Differential Calculus | "Limits and continuity", "Derivatives: definition and basic rules", "Derivatives: chain rule and other advanced topics", "Applications of derivatives", "Analyzing functions", "Parametric equations, polar coordinates, and vector-va", |
| Integral Calculus | "Integrals", "Differential equations", "Applications of integrals", "Parametric equations, polar coordinates, and vector-valued functions", "Series", |
| AP College Calculus AB | "Limits and continuity", "Differentiation: definition and basic derivative rules", "Differentiation: composite, implicit, and inverse functions", "Contextual applications of differentiation", "Applying derivatives to analyze functions", "Integration and accumulation of change", "Differential equations", "Applications of integration", "AP Calculus AB solved free response questions from past exams", "AP Calculus AB Standards mappings", |
| AP College Calculus BC | "Limits and continuity", "Differentiation: definition and basic derivative rules", "Differentiation: composite, implicit, and inverse functions", "Contextual applications of differentiation", "Applying derivatives to analyze functions", "Integration and accumulation of change", "Differential equations", "Applications of integration", "Parametric equations, polar coordinates, and vector-valued functions", "Infinite sequences and series", "AP Calculus BC solved exams", "AP Calculus BC Standards mappings", |
| Multivariable calculus | "Thinking about multivariable functions", "Derivatives of multivariable functions", "Applications of multivariable derivatives", "Integrating multivariable functions", "Green's, Stokes', and the divergence theorems", |
| Differential equations | "First order differential equations", "Second order linear equations", "Laplace transform", |
| Linear algebra | "Vectors and spaces", "Matrix transformations", "Alternate coordinate systems (bases)", |

points with 2 seed concepts, more than 90% of the data can not pass the prover validation at the first generation. And almost 30% of them cost 2 correction steps to obtain valid proof. Similar

Table 13: Prompt template for proof filtering.

---

**Prompt Template for Proof Filtering**

```
In the following, you are given a ''Problem'', a pair of corresponding
''Informal proof'' and ''Formal proof in Lean 3'', along with error
messages from a Lean Prover corresponding to the ''Formal proof in Lean
3''.  Now please carefully modify the ''Formal proof in Lean 3'' section
so that it passes the Lean Prover without error.  You should write the
modified complete proof in your response.

# Problem:  <generated problem>

# Informal proof:  <generated informal proof>

# Formal proof (1) in Lean 3:
'''lean
line 1 
line 2 
line 3 
...
'''

# Error messages for Formal proof (1) from Lean Prover:
<error messages>

...

# Formal proof (k) in Lean 3:
'''lean
line 1 
line 2 
line 3 
...
'''

# Error messages for Formal proof (k) from Lean Prover:
<error messages>
```

---

observations are found in word problems with 2 seed concepts. It is suggested that the data subset with 2 seed concepts is challenging to the LLMs in general. In contrast, data with 1 seed concept are easier for LLMs. But there are still more than half data points that need proof improvements based on error messages from the theorem prover. Therefore, overall, MUSTARD generates valid data points in different difficulty levels, and the majority of the problems are challenging for the LLMs.

### D.2 DATA DIVERSITY

We compute ROUGE-L (Lin, 2004) to check the diversity of generated informal statements and proofs. Specifically, given a data set, we perform 10 rounds of bootstrapping. In each round, we randomly sample 10 data points from the data set, each of which is paired with the remaining data points, and compute pair-wise ROUGE-L scores. The ROUGE-L score per round is obtained by averaging the pair-wise scores. The final ROUGE-L score is an average score over the bootstrapping. We compare the scores among the generation settings, and the results are shown in Figure 6.

The results show that all settings have a ROUGE-L score beneath 0.25, which indicates a high diversity of the generated informal statements and informal proofs. All-at-once and step-by-step generation share similar data diversity. The ROUGE-L scores slightly increase over the educational levels. Generating higher-educational theorem proving data with 2 concepts step-by-step exceeds 0.2 in ROUGE-L. Therefore, this setting is relatively challenging.

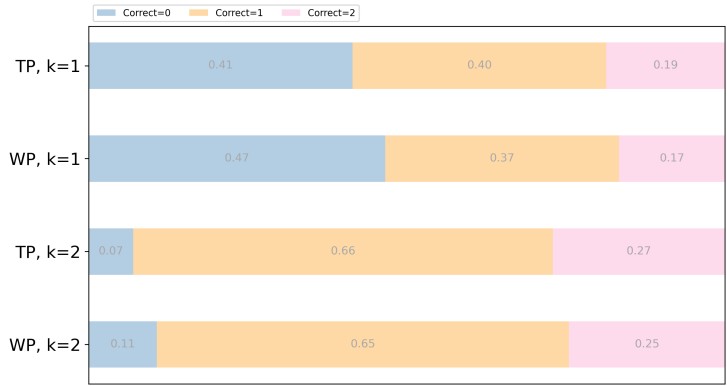

Figure 5: Data proportions according to correction steps.

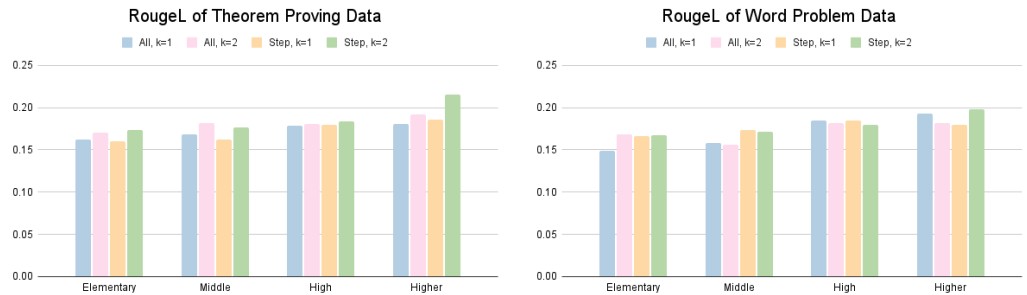

Figure 6: Diversity of informal statements and proofs by ROUGE. All: All-at-once generation. Step: Step-by-step generation. k: Number of seed concepts.

We further investigate the diversity of formal statements and proofs. We collect all the occurrences of used tactics and lemmas, and Figure 7 shows the distributions. It is demonstrated that LLMs tend to use diverse lemmas to solve the problems. And LLMs memorize and understand multiple lemmas. The most commonly used lemmas in both theorem proving and word problems include `of_as_true`, `mul_comm`, and `nat.mul_comm`. `of_at_true` is often used in automated proofs for complex propositions to first prove their decidability. `mul_comm` and `nat.mul_comm` are about the multiplicative commutative law and are often used for expression simplification. Similarly, the frequently used `neg_pos`, `pow_two`, `nat.div_eq_of_eq_mul_right` and so forth are basic lemmas for proving advanced properties. Similar observations are found in tactics. Therefore, LLMs are good at breaking problems into basic steps and using basic lemmas to complete complex proving.

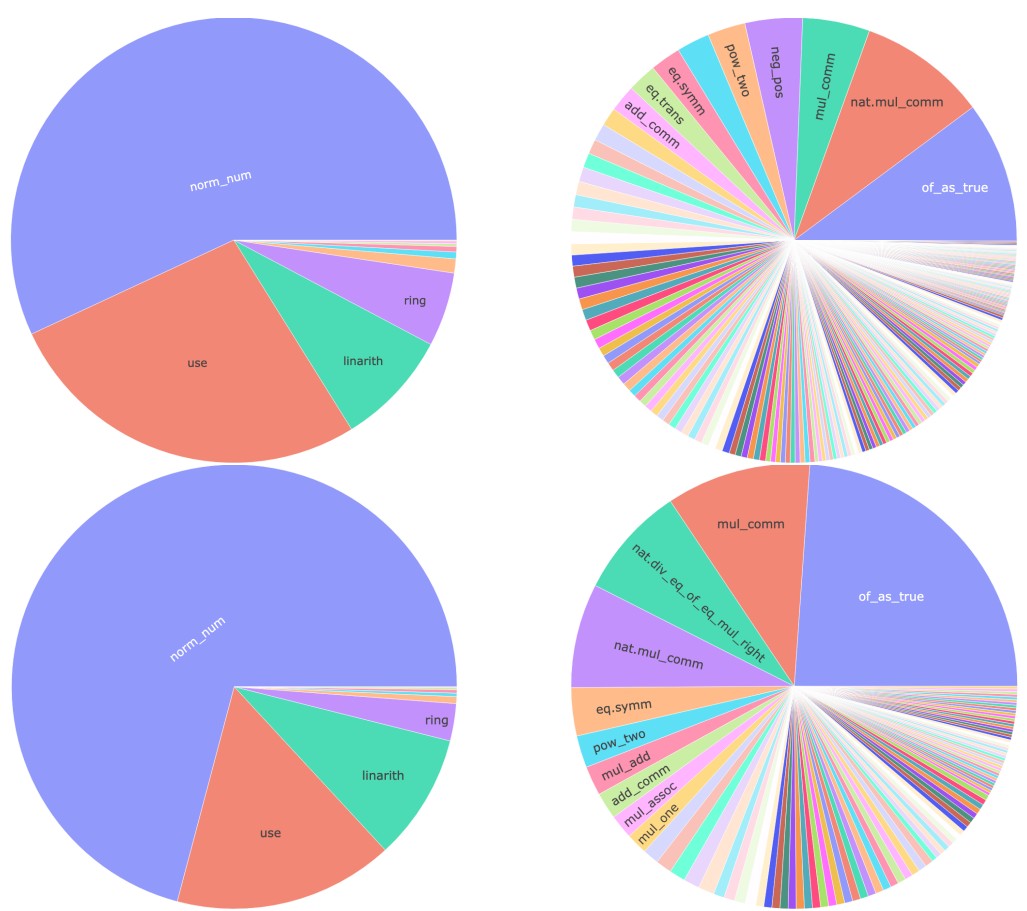

Figure 7: Distribution of tactics (left-hand side) and lemmas (right-hand side) used by the formal statements and proofs in theorem proving (upper row) and word problem (lower row).

# E CASE STUDY

Tables 15 to 26 demonstrates successful and failed cases generated by MUSTARD.

## F    IMPLEMENTATION DETAILS OF DOWNSTREAM TASK

### F.1    DATASETS

**GSM8K (Cobbe et al., 2021)**    GSM8K consists of 8.5K elementary mathematics word problems that require 2 to 8 arithmetic operations to arrive at the final answer. The dataset comprises 7.5K training questions and 1K test questions. Inspired by Kojima et al. (2022), during inference, we use appropriate prompts and examples to prompt the model for zero-shot and few-shot reasoning. The used prompts are demonstrated in Table 27.

**Mathlib**    Mathlib[6] is a community-maintained library designed for the Lean theorem prover. It encompasses both programming tools and mathematical content, along with tactics that leverage these tools to facilitate mathematical development. The version of Mathlib we use is consistent with Wang et al. (2023a). The lengths of the training, test, and validation sets were 36,960, 1,621, and 1,580, respectively.

**miniF2F (Zheng et al., 2022)**    MiniF2F serves as a formal mathematics benchmark, which has been translated to work with multiple formal systems. It encompasses exercise statements from olympiads like AMC, AIME, and IMO, in addition to content from high-school and undergraduate mathematics courses. The MiniF2F test split contains 244 formal Olympiad-level mathematics problem statements. We use the auto-regression method to construct the training corpus like Polu & Sutskever (2020); Han et al. (2022); Wang et al. (2023a). We perform an evaluation using best-first search with a number of expansions per proof search of $d = 8$ during inference.

### F.2    MODELS

**GPT2-large (Radford et al., 2019)**    The GPT2-large model is a transformer language model, following the decoder-only architecture introduced by Vaswani et al. (2017). The model boasts an impressive 774 million parameters, 36 layers, 20 attention heads, and a hidden dimension of 1,280. Additionally, it employs a tokenizer featuring a vocabulary size of 50,400. The model is pre-trained on Github python codes and the arXiv library.

**Llama 2-7B (Touvron et al., 2023)**    Llama 2 is a language model that employs an auto-regressive transformer architecture, pre-trained on open-source corpus. The model utilizes both supervised fine-tuning and reinforcement learning with human feedback techniques to align with human preferences. The 7B model is configured with 32 layers, 32 attention heads, and a hidden dimension of 4,096.

### F.3    IMPLEMENTATION DETAILS

We employ LoRA (Hu et al., 2021) for fine-tuning the pre-trained models on MUSTARDSAUCE, where the trainable parameters of GPT2-large and Llama 2-7B constitute 19% and 6%, respectively. The training is conducted with a maximum of 10 epochs using a batch size of 16 and a warm-up step of 1,000, with a maximum learning rate of 1e-4 and a minimum learning rate of 5e-6. The best checkpoint is selected based on the minimum perplexity of the validation split.

## G    MORE EXPERIMENTAL RESULTS

Table 28 demonstrates the compared results on GSM8K and MATH between Llama 2-7B and Llama 2-70B. Llama 2-70B fine-tuned with MUSTARDSAUCE-valid consistently outperforms the model fine-tuned with MUSTARDSAUCE-random by 8.33% in the zero-shot manner and 5.30% in the few-shot manner. It also surpasses the model fine-tuned with the invalid subset and the entire generated dataset. The results also suggest the effectiveness of the framework with a larger fine-tuned LM.

---

[6]https://github.com/leanprover-community/mathlib

## H    DATA CONTAMINATION CHECK

We check cross-contamination between MUSTARDSAUCE and the evaluation datasets inspired by OpenAI (2023). However, instead of using a substring match that may result in false negatives and false positives, we compute cosine similarities based on text-embedding-ada-002[7], and then inspect the nearest data points in the paired datasets. The automated theorem proving (ATP) dataset miniF2F only contains formal statements and proofs, while the math word problem (MWP) dataset GSM8K only contains informal statements and proofs. For a more detailed inspection, we check the corresponding fractions in MUSTARDSAUCE. Tables 29, 32, 31, and 30 demonstrate the inspected cases. The nearest data points are dissimilar. Therefore, we exclude data contamination of the generated MUSTARDSAUCE according to these observations.

---

[7]https://openai.com/blog/new-and-improved-embedding-model

Table 14: Prompt templates for step-by-step generation in the proof-generation stage.

**Prompt Templates for Step-by-Step Generation**

(T1)
```
You are a math expert.  Now please come up with a math problem
according to the following requirements.  The math problem should
contain a question part (indicated by ``Problem:  ''), a
corresponding solution in natural language (indicated by ``Informal
proof:''), and a translated formal solution in Lean 3 (indicated by
``Formal proof in Lean 3:'').  Please note that the informal proof
and the formal proof need to be identical.  Please create a
[QUESTION TYPE] in the level of [EDUCATIONAL LEVEL] based on the
following knowledge point(s):  [CONCEPT] in [DOMAIN]; [CONCEPT] in
[DOMAIN].

Please first write the question part regardless of the other parts.
You must write the following format, filling in the ``# Problem:
''  section, and leaving the other two sections empty.

# Problem:  ...

# Informal proof:  ...

# Formal proof in Lean 3:  ...
```

(T2)
```
You are a math expert.  Now please come up with a math problem
according to the following requirements.  The math problem should
contain a question part (indicated by ``Problem:  ''), a
corresponding solution in natural language (indicated by ``Informal
proof:''), and a translated formal solution in Lean 3 (indicated by
``Formal proof in Lean 3:'').  Please note that the informal proof
and the formal proof need to be identical.  Please create a
[QUESTION TYPE] in the level of [EDUCATIONAL LEVEL] based on the
following knowledge point(s):  [CONCEPT] in [DOMAIN]; [CONCEPT] in
[DOMAIN].

Please then write the corresponding solution in natural language
(indicated by ``Informal proof:'')  given the ``# Problem:  '',
filling in the ``# Informal proof:  ''  section, and leaving the
other section empty.

# Problem:  <generated problem>

# Informal proof:  ...

# Formal proof in Lean 3:  ...
```

(T3)
```
You are a master in Lean.  Now please come up with a math problem
according to the following requirements.  The math problem should
contain a question part (indicated by ``Problem:  ''), a
corresponding solution in natural language (indicated by ``Informal
proof:''), and a translated formal solution in Lean 3 (indicated by
``Formal proof in Lean 3:'').  Please note that the informal proof
and the formal proof need to be identical.  Please create a
[QUESTION TYPE] in the level of [EDUCATIONAL LEVEL] based on the
following knowledge point(s):  [CONCEPT] in [DOMAIN]; [CONCEPT] in
[DOMAIN].

Please translate the ``# Informal proof:''  section into Lean 3 and
fill in the ``# Formal proof in Lean 3:  ''  section.

# Problem:  <generated problem>

# Informal proof:  <generated informal proof>

# Formal proof in Lean 3:  ...
```

Table 15: A successful case generated by MUSTARD.

| **Question Type: Word Problem.** **Educational Level: Elementary School.** **k=1.** **Successful case.** |
|---|
| **Concept(s):** *Division* in 4th grade. |
| **Comments:** MUSTARD accurately counts the number of people and correctly performs division. |

**Informal Statement**
John and his 3 friends are on a trip. They bought a huge box of chocolates containing 240 pieces. They decided to split the chocolates equally among themselves. How many chocolates will each one of them get?

**Informal Proof**
Since there are 4 people in total and they have 240 chocolates, we can find out how many chocolates each person will get by dividing the total number of chocolates by the total number of people. So, 240 divided by 4 equals 60. Therefore, each person will get 60 chocolates.

**Formal Statement and Proof**
```
def divide (x y :  ℕ) :  ℕ := x / y

#reduce divide 240 4 -- The output should be 60, which means each person
will get 60 chocolates.
```

Table 16: A failed cased generated by MUSTARD.

| **Question Type: Theorem Proving.** **Educational Level: Elementary School.** **k=1.** **Failed case.** |
|---|
| **Concept(s):** *Quadrilaterals* in 3rd grade. |
| **Comments:** MUSTARD believes that a quadrilateral with four equal sides is a square. It treats a necessary condition as a sufficient condition. |

**Informal Statement**
Given a quadrilateral ABCD where AB = BC = CD = DA = 5 units, prove that this quadrilateral is a square.

**Informal Proof**
The given quadrilateral ABCD has all its sides equal. This is a property of a square. Hence, ABCD is a square.

**Formal Statement and Proof**
```
import data.real.basic

def is_square_quad (AB BC CD DA : ℝ) :  Prop := AB = BC ∧ BC = CD ∧ CD =
DA

def ABCD_is_square :  ∀ (AB BC CD DA : ℝ), AB = 5 ∧ BC = 5 ∧ CD = 5 ∧ DA
= 5 → is_square_quad AB BC CD DA :=
begin
 intros AB BC CD DA h,
 cases h with h₁ h₂,
 cases h₂ with h₃ h₄,
 cases h₄ with h₅ h₆,
 split,
 { rw h₁, exact h₃ },
 split,
 { rw h₃, exact h₅ },
 { rw h₅, exact h₆ },
end
```

Table 17: A successful case generated by MUSTARD.

| Question Type: Word Problem.   Educational Level: Elementary School.   k=2. | Successful case. |
|---|---|

**Concept(s):** *Addition and subtraction* in 1st grade; Measuring angles in 4th grade.

**Comments:** MUSTARD creatively combines the two concepts and constructs a reasonable question.

**Informal Statement**
John has a favorite toy car that he likes to move around a circular track. He starts moving the car from a certain point and stops after the car has moved 90 degrees. Then he moves the car again but this time in the opposite direction, stopping when the car has moved 45 degrees. What is the net degree movement of John's toy car?

**Informal Proof**
Initially, John moves his car 90 degrees. Then he moves the car in the opposite direction 45 degrees. Since the car is moving in the opposite direction, we consider this as -45 degrees. To find the net movement, we add the two movements together. So, 90 + (-45) equals 45. Therefore, the net degree movement of John's toy car is 45 degrees.

**Formal Statement and Proof**
```
import data.int.basic

open int

-- define the initial movement
def initial_movement :  ℤ := 90

-- define the backward movement
def backward_movement :  ℤ := -45

-- define the function to calculate the net movement
def net_movement (initial backward :  ℤ) :  ℤ := initial + backward

-- proof that the net movement is 45 degrees
example :  net_movement initial_movement backward_movement = 45 :=
begin
 unfold net_movement,
 unfold initial_movement,
 unfold backward_movement,
 exact dec_trivial,
end
```

Table 18: A successful case generated by MUSTARD.

| Question Type: Theorem Proving.   Educational Level: Middle School.   k=2. | Successful case. |
| --- | --- |

**Concept(s):** *Geometry* in 8th grade; *Algebraic expressions* in Algebra basics.

**Comments:** MUSTARD understands polynomials and writes a valid proof in Lean.

**Informal Statement**
Given a rectangle ABCD where AB is x + 5 and AD is 2x - 3. Prove that the area of the rectangle ABCD is $(2x^2 + 7x - 15)$ square units.

**Informal Proof**
In a rectangle, the area is calculated by multiplying the length and the width. Here, the length is represented by the expression $x + 5$ and the width is represented by the expression $2x - 3$. By multiplying these two expressions together, we get:

$$(x + 5) * (2x - 3) = 2x^2 - 3x + 10x - 15 = 2x^2 + 7x - 15$$

Therefore, the area of the rectangle ABCD is $(2x^2 + 7x - 15)$ square units.

**Formal Statement and Proof**
```
import algebra.ring.basic

variables (x : ℤ)

def rectangle_area :  (x + 5) * (2 * x - 3) = 2 * (x * x) + 7 * x - 15 :=
begin
 have h₁ :  (x + 5) * (2 * x - 3) = 2 * (x * x) - 3 * x + 10 * x - 15, by
ring,
 have h₂ :  2 * (x * x) - 3 * x + 10 * x - 15 = 2 * (x * x) + 7 * x - 15,
by simp,
 rw h₁,
 exact h₂,
end
```

Table 19: A failed case generated by MUSTARD.

| **Question Type: Theorem Proving.**  **Educational Level: High School.**  **k=1.**  **Failed case.** |
| --- |
| **Concept(s):** *Displaying and comparing quantitative data* in Statistics and probability. |
| **Comments:** MUSTARD gives the definition of median in the formal proof. |

**Informal Statement**
Given two sets of data A = {5, 7, 10, 12, 15} and B = {6, 8, 11, 14, 16}, we are to find the median of the two sets and prove that the median of set A is less than the median of set B.

**Informal Proof**
To find the median of a given data set, we need to sort the data in ascending order and then find the middle value. In the case of set A and set B, both are already sorted in ascending order. For set A, the median is 10 and for set B, the median is 11. Therefore, it is clear that the median of set A is less than the median of set B.

**Formal Statement and Proof**
```
import data.set.finite
import data.set.lattice

def A : finset ℕ := 5, 7, 10, 12, 15.to_finset
def B : finset ℕ := 6, 8, 11, 14, 16.to_finset

def median (A : finset ℕ) :  ℕ :=
if A.card % 2 = 1 then (A.sort (≤)).get ((A.card - 1) / 2) else
(A.sort (≤)).get (A.card / 2 - 1) / 2 + (A.sort (≤)).get (A.card / 2) /
2

lemma median_A_less_than_B :
 (median A) < (median B) :=
begin
 simp [median, A, B, finset.sort, finset.get],
 norm_num,
end
```

Table 20: A failed case generated by MUSTARD.

| | | | |
|---|---|---|---|
| **Question Type: Word Problem.** | **Educational Level: High School.** | **k=2.** | **Failed case.** |

**Concept(s):** *Integrals* in Calculus 1; *Summarizing quantitative data* in Statistics and probability.

**Comments:** The informal statement does not cover the situation of leap year. The solution does not give the final answer.

**Informal Statement**

A company produced a new model of a car and started selling it on January 1. The number of cars sold $t$ days after January 1 is given by the function $f(t) = 2t^2 + 3t + 1$. Use integrals to find the total number of cars sold from January 1 to March 31. Then calculate the average number of cars sold per day during this period.

**Informal Proof**

The total number of cars sold from January 1 to March 31 can be found by integrating the function $f(t)$ from $t = 0$ (January 1) to $t = 90$ (March 31), as each day corresponds to one unit of $t$.

Performing the integration, we get $\int (2t^2 + 3t + 1) \, \mathrm{d}t$ from 0 to 90, which equals $[\frac{2}{3} \times t^3 + \frac{3}{2} \times t^2 + t]$ from 0 to 90. Substituting $t = 90$ into the integrated function and subtracting the value at $t = 0$, we get the total number of cars sold.

The average number of cars sold per day is then the total number of cars sold divided by 90 (the number of days from January 1 to March 31).

**Formal Statement and Proof**
```
import data.real.basic

open real

def f := λ(t : ℝ), 2*t² + 3*t + 1

lemma f_value_at_90 :  f 90 = 2*(90)² + 3*(90) + 1 :=
begin
 simp [f],
end
```

Table 21: A successful case generated by MUSTARD.

| Question Type: Theorem Proving. Educational Level: Higher Education. k=1. | Successful case. |
|---|---|

**Concept(s):** *Alternate coordinate systems (bases)* in Linear algebra.

**Comments:** MUSTARD correctly applies the lemma. All the concepts of linear algebra are found.

**Informal Statement**

Let $V$ be a vector space, and let $B_1 = v_1, v_2, ..., v_n$ and $B_2 = w_1, w_2, ..., w_n$ be two bases for $V$. Prove that there exists an invertible linear map $T : V \to V$ such that $T(v_i) = w_i$ for each $i$ in $1, 2, ..., n$.

**Informal Proof**

The map $T : V \to V$ defined by $T(v_i) = w_i$ for each $i$ is linear: indeed, for any vectors $v$, $v'$ in $B_1$ and any scalars $c$, $c'$, we have $T(cv + c'v') = cw + c'w'$ where $w = T(v)$ and $w' = T(v')$. This follows from the linearity properties of vector addition and scalar multiplication in the vector space $V$.

**Formal Statement and Proof**

```
import linear_algebra.basis

variables V : Type* [add_comm_group V] [module ℝ V]
variables v w :  V
variables B1 B2 :  set V
variables (hB1 :  is_basis ℝ B1) (hB2 :  is_basis ℝ B2)

lemma exists_invertible_linear_map :
 ∃ (T : V →[ℝ] V) (hT : is_linear_map ℝ T) (hinv :  function.bijective
T),
 ∀ (v ∈ B1), T v ∈ B2 :=
begin use [hB2.constr hB1, hB2.is_linear hB1, hB2.bijective hB1],
 intros v hv,
 apply hB2.mem_span,
 rw hB1.repr_self hv,
 exact λ _, hB2.smul _ _,
end
```

Table 22: A successful case generated by MUSTARD.

| Question Type: Word Problem. Educational Level: Higher Education. k=1. Successful case. |
| --- |
| **Concept(s):** *Series* in Integral Calculus. |
| **Comments:** MUSTARD is proficient in number series and is able to perform detailed proof by integration. |

**Informal Statement**

Prove that the integral of the series $\sum$ (from $n = 1$ to infinity) of $\frac{-1^{(n+1)}}{n}$ equals $\ln(2)$.

**Informal Proof**

We start by recognizing the series $\sum$ (from $n = 1$ to infinity) of $\frac{-1^{(n+1)}}{n}$ as the alternating harmonic series, which is known to converge to $\ln(2)$.

**Formal Statement and Proof**

```
import data.real.basic
import analysis.special_functions.pow

open real

lemma integral_of_series :
  ∫ x in 0..1, (ln (1 + x)) = ∑' (n :  ℕ), ((-1)^(n+1) / (n + 1)) :=
begin
 have h_der :  ∀ x ∈ set.Icc (0 :  ℝ) 1, deriv (λ x, (ln (1 + x))) x = 1
/ (1 + x),
 { intros x hx,
 have h_pos :  0 < 1 + x := by nlinarith,
 exact deriv_ln_one_add h_pos },

 have h_int :  interval_integral (λ x, 1 / (1 + x)) 0 1 volume = ∑' (n :
ℕ), ((-1)^(n+1) / (n + 1)),
 { have h_frac :  ∀ (n :  ℕ), ∫ x in 0..1, x^n = 1 / (n + 1),
 { intro n,
 calc ∫ x in 0..1, x^n = [x^(n+1) / (n+1)] | 0..1 :  integral_pow n
 ...  = 1 / (n + 1) :  by { rw integral_interval, simp } },
 rw [interval_integral.integral_of_le, h_frac],
 simp,
 linarith },

 have h_eq :  ∫ x in 0..1, (ln (1 + x)) = ∫ x in 0..1, 1 / (1 + x),
 { congr, ext, exact h_der x (set.mem_Icc.mpr ⟨by nlinarith, by
nlinarith⟩) },

 rw [h_eq, h_int],
end
```

Table 23: A failed case generated by MUSTARD.

| Question Type: Theorem Proving. Educational Level: Higher Education. k=2. Failed case. |
| --- |

**Concept(s):** *Second order linear equations* in Differential equations; *Limits and continuity* in AP College Calculus BC.

**Comments:** MUSTARD does not understand second-order linear equations and continuity.

**Informal Statement**
Given a second order linear differential equation of the form $y'' + p(x)y' + q(x)y = 0$, where $p(x)$ and $q(x)$ are continuous on the interval $[a, b]$, prove that if there exist two solutions $y1$ and $y2$ that are linearly independent, then any solution y of the differential equation can be expressed as $y = c1 * y1 + c2 * y2$, where $c1$ and $c2$ are constants.

**Informal Proof**
The general solution to a homogeneous second order linear differential equation is given by the superposition of two linearly independent solutions. This is a consequence of the linearity of the differential operator.

Suppose we have a second order linear differential equation, and $y1$ and $y2$ are two solutions that are linearly independent. Then, any solution $y$ of the differential equation can be expressed as a linear combination of $y1$ and $y2$. This means there exist constants $c1$ and $c2$ such that $y = c1 * y1 + c2 * y2$.

The reason is that any linear combination of solutions to a linear differential equation is also a solution to the differential equation. This follows from the properties of the derivative, which allow us to distribute the derivative over addition and to factor out constants.

**Formal Statement and Proof**
```
import data.real.basic

variables α :  Type* [field α]

/-- Placeholder for the differential equation -/
def diff_eq (p q y :  α → α) (x :  α) :  Prop :=
y x = 0 -- This is just a placeholder.  A real differential equation
would have a more complex form.

/-- Theorem:  Given a second order linear differential equation of the
form y'' + p(x)y' + q(x)y = 0,
where p(x) and q(x) are continuous on the interval [a, b], if there exist
two solutions y1 and y2
that are linearly independent, then any solution y of the differential
equation can be expressed
as y = c1*y1 + c2*y2, where c1 and c2 are constants.  -/
lemma second_order_linear_differential_eq (p q y1 y2 y :  α → α) (c1 c2 :
α) :
(∃ x, diff_eq p q y1 x ∧ diff_eq p q y2 x) → (∀ x, diff_eq p q y x) →
(∀ x, y x = c1 * y1 x + c2 * y2 x) :=
begin
 intros h_sol h_y x,
 -- This is a placeholder.  A real proof would show that y x can be
written as a linear combination
 -- of y1 x and y2 x using the properties of the differential equation,
the assumption that y1 and y2
 -- are solutions, and the linearity of the differential operator.
 sorry
end
```

Table 24: A successful case generated by MUSTARD.

| Question Type: Word Problem. Educational Level: Higher Education. k=2. | Successful case. |
|---|---|

**Concept(s):** *Matrix transformations* in Linearalgebra; *Collecting data* in AP College Statistics.

**Comments:** MUSTARD performs correct matrix transformation.

**Informal Statement**
Consider two matrices $A = [1, 2; 3, 4]$ and $B = [5, 6; 7, 8]$ which represent two different data collection methods applied to a survey of a population. We want to find the collective effect of these two data collection methods when applied simultaneously. In other words, find the matrix $C$ which is the result of the matrix transformation of $A$ by $B$.

**Informal Proof**
Matrix multiplication is a form of transformation of one matrix by another. In this case, we want to find the matrix $C$ that is a result of the transformation of $A$ by $B$.

Matrix multiplication is done by taking the dot product of the rows of the first matrix with the columns of the second matrix.

Therefore,
$C[0, 0] = (A[0, 0] * B[0, 0]) + (A[0, 1] * B[1, 0]) = (1 * 5) + (2 * 7) = 19$,
$C[0, 1] = (A[0, 0] * B[0, 1]) + (A[0, 1] * B[1, 1]) = (1 * 6) + (2 * 8) = 22$,
$C[1, 0] = (A[1, 0] * B[0, 0]) + (A[1, 1] * B[1, 0]) = (3 * 5) + (4 * 7) = 43$, and
$C[1, 1] = (A[1, 0] * B[0, 1]) + (A[1, 1] * B[1, 1]) = (3 * 6) + (4 * 8) = 50$.

So, the matrix $C = [19, 22; 43, 50]$.

**Formal Statement and Proof**
```
import data.matrix.basic

def A : matrix (fin 2) (fin 2) ℕ := !![![1, 2], ![3, 4]]
def B : matrix (fin 2) (fin 2) ℕ := !![![5, 6], ![7, 8]]
def C : matrix (fin 2) (fin 2) ℕ := A.mul B

#eval C
-- Output:  !![![19, 22], ![43, 50]]
```

Table 25: A failed case generated by MUSTARD.

---

**Question Type: Theorem Proving.   Educational Level: Higher Education.   k=2.   Failed case.**

**Concept(s):** *Green's, Stokes', and the divergence theorems* in Multivariable calculus; *Differential equations* in Integral Calculus.

**Comments:** MUSTARD does not finish the formal proof with `sorry`.

**Informal Statement**
Let $F$ be a continuously differentiable vector field on $\mathbb{R}^3$. Let $S$ be the boundary of the unit ball $B$ in $\mathbb{R}^e$, oriented by the outward normal. Let $n$ denote the outward unit normal on $S$. Prove that $\int S$ div $F \, \mathrm{d}S = \oint SF \bullet n \, \mathrm{d}S$, where "div" denote the divergence of $F$, "$\bullet$" denotes the dot product, and "$\int$", "$\oint$" denote the surface integral and line integral respectively.

**Informal Proof**
This is an application of the divergence theorem, also known as Gauss's theorem.

The divergence theorem states that the surface integral of a vector field over a surface $S$ is equal to the volume integral of the divergence of the field over the volume $V$ enclosed by $S$. In mathematical terms, $\int SF \bullet n \, \mathrm{d}S = \oint V$ div $F \, \mathrm{d}V$ .

In our case, the volume $V$ is the unit ball $B$ in $\mathbb{R}^3$, and the surface $S$ is the boundary of $B$. The divergence theorem thus gives us $\int SF \bullet n \, \mathrm{d}S = \oint B$ div $F \, \mathrm{d}V$ .

However, since $B$ is the unit ball, its volume is 1, so the volume integral over $B$ is just the average value of div $F$ over $B$. Therefore, $\oint B$ div $F \, \mathrm{d}V = \int S$ div $F \, \mathrm{d}S$ .

Putting everything together, we obtain $\int SF \bullet n \, \mathrm{d}S = \int S$ div $F \, \mathrm{d}S$ , as required.

**Formal Statement and Proof**
```
import analysis.calculus.fderiv
import measure_theory.integration
import analysis.normed_space.inner_product
import data.real.basic
import topology.instances.real
import analysis.special_functions.trigonometric

open_locale classical

noncomputable theory

/- Let's define ℝ³ as Euclidean space -/
def ℝ³ := euclidean_space ℝ (fin 3)

/- Define the ball and boundary -/

def B := metric.ball (0:ℝ³) 1
def S := metric.sphere (0:ℝ³) 1

/- Assume F is a continuously differentiable vector field on ℝ³ -/
variables (F : ℝ³ → ℝ³) (hF : ∀ x, differentiable_at ℝ F x)

/- Define the divergence of F -/
def divF := λ x, (fderiv ℝ F x).trace

/- Define the outward unit normal on S -/
def n := λ x:ℝ³, x

/- Now we can state the theorem -/
def div_eq_surface_integral :
 ∫ x in B, divF F x = ∮ x in S, inner (F x) (n x) :=
begin
-- This is where the main proof would go, using the divergence theorem
and other facts about integration.
 sorry
end
```

Table 26: A failed case generated by MUSTARD.

| **Question Type: Theorem Proving.** **Educational Level: Higher Education.** **k=2.** **Failed case.** |
|---|

**Concept(s):** *Matrix transformations* in Linear algebra.

**Comments:** The generated informal statement has factual errors.

**Informal Statement**
Let's consider a function $f : \mathbb{R} \to \mathbb{R}$ that is differentiable at a point $x_0$. Let's also consider a $2 \times 2$ matrix $A$. We define a new function $g(x) = A * f(x)$. Prove that $g$ is also differentiable at $x_0$, and find the derivative of $g$ at $x_0$. **Informal Proof**
Since $f$ is differentiable at $x_0$, the limit of the difference quotient exists at this point. Denote this limit as $f'(x_0)$.

We know that matrix multiplication is a linear operation. So we can move the limit inside the matrix multiplication to find the derivative of $g$ at $x_0$.

So, $g'(x_0) = A * f'(x_0)$.

**Formal Statement and Proof**
```
import data.matrix.basic
import analysis.calculus.deriv

variables f :  ℝ → ℝ A : matrix (fin 2) (fin 2) ℝ x₀ :  ℝ
variable (h :  differentiable_at ℝ f x₀)

include h

lemma matrix_times_differentiable_function :  differentiable_at ℝ (λ x, A ·
(λ i, f x) i) x₀ :=
begin
 apply matrix.has_fderiv_at_diff_at,
 exact h.has_deriv_at,
end

lemma derivative_of_matrix_times_function :  deriv (λ x, A · (λ i, f x) i)
x₀ = A · (λ i, deriv f x₀) i :=
begin
 apply has_deriv_at.deriv,
 exact matrix_times_differentiable_function h,
end
```

Table 27: Prompt template for math word problem inference.

---

**Zero shot prompt template for math word problem inference**

---

```
You are an expert in math.  Answer the following math word problem.
Question:  <question>
Answer:  Let's think step by step.
```

---

**Few shot prompt template for math word problem inference**

---

```
You are an expert in math.  Answer the following math word problem.

Question:  There are 15 trees in the grove.  Grove workers will plant
trees in the grove today.  After they are done, there will be 21 trees.
How many trees did the grove workers plant today?
Answer:  There are 15 trees originally.  Then there were 21 trees after
some more were planted.  So there must have been 21 - 15 = 6.  The answer
is 6.

Question:  If there are 3 cars in the parking lot and 2 more cars arrive,
how many cars are in the parking lot?
Answer:  There are originally 3 cars.  2 more cars arrive.  3 + 2 = 5.
The answer is 5.

Question:  Leah had 32 chocolates and her sister had 42.  If they ate 35,
how many pieces do they have left in total?
Answer:  Originally, Leah had 32 chocolates.  Her sister had 42.  So in
total they had 32 + 42 = 74.  After eating 35, they had 74 - 35 = 39.
The answer is 39.

Question:  Jason had 20 lollipops.  He gave Denny some lollipops.  Now
Jason has 12 lollipops.  How many lollipops did Jason give to Denny?
Answer:  Jason started with 20 lollipops.  Then he had 12 after giving
some to Denny.  So he gave Denny 20 - 12 = 8.  The answer is 8.

Question:  Shawn has five toys.  For Christmas, he got two toys each from
his mom and dad.  How many toys does he have now?
Answer:  Shawn started with 5 toys.  If he got 2 toys each from his mom
and dad, then that is 4 more toys.  5 + 4 = 9.  The answer is 9.

Question:  There were nine computers in the server room.  Five more
computers were installed each day, from monday to thursday.  How many
computers are now in the server room?
Answer:  There were originally 9 computers.  For each of 4 days, 5 more
computers were added.  So 5 * 4 = 20 computers were added.  9 + 20 is 29.
The answer is 29.

Question:  Michael had 58 golf balls.  On tuesday, he lost 23 golf balls.
On wednesday, he lost 2 more.  How many golf balls did he have at the end
of wednesday?
Answer:  Michael started with 58 golf balls.  After losing 23 on tuesday,
he had 58 - 23 = 35.  After losing 2 more, he had 35 - 2 = 33 golf balls.
The answer is 33.

Question:  Olivia has $23.  She bought five bagels for $3 each.  How much
money does she have left?
Answer:  Olivia had 23 dollars.  5 bagels for 3 dollars each will be 5
x 3 = 15 dollars.  So she has 23 - 15 dollars left.  23 - 15 is 8.  The
answer is 8.

Question:  <question>
Answer:
```

---

Table 28: Compared performances on GSM8K (G) and MATH (M) between Llama 2-7B and Llama 2-70B.

| MODEL | Zero (G) | Few (G) | Zero (M) | Few (M) |
|---|---|---|---|---|
| *Baselines* | | | | |
| Llama 2-7B | 7.2 | 12.8 | 2.0 | 2.6 |
| Llama 2-70B | 31.7 | 54.1 | 8.8 | 13.4 |
| *Fine-tuning* | | | | |
| Llama 2-7B > tt | 9.6 | 16.0 | 3.2 | 3.8 |
| Llama 2-7B > in | 9.1 | 14.9 | 2.4 | 3.2 |
| Llama 2-7B > ra | 9.5 | 15.4 | 3.0 | 3.6 |
| Llama 2-7B > va | 10.3 (+8.42%) | 16.9 (+9.74%) | 3.2 (+6.67%) | 4.2 (+16.67%) |
| Llama 2-70B > tt | 36.6 | 55.8 | 10.0 | 14.4 |
| Llama 2-70B > in | 33.4 | 53.7 | 9.2 | 13.6 |
| Llama 2-70B > ra | 36.1 | 55.4 | 9.6 | 14.2 |
| Llama 2-70B > va | 39.5 (+9.42%) | 59.1 (+6.68%) | 10.4 (+8.33%) | 15.0 (+5.30%) |

Table 29: Nearest data points between MUSTARDSAUCE and miniF2F.

---

**MUSTARDSAUCE v.s. miniF2F (cosine similarity = 0.6439)**

---

**MUSTARDSAUCE Case**

**Informal Statement:**

Alex has 5 ten-dollar bills and 3 one-dollar bills. How much money does Alex have in total?

**Informal Proof:**

To find out how much money Alex has in total, we need to multiply the number of each type of bill by its value. So, Alex has 5 ten-dollar bills, which equals 5 * 10 = 50 dollars. He also has 3 one-dollar bills, which equals 3 * 1 = 3 dollars. Adding these two amounts together gives 50 + 3 = 53 dollars. Therefore, Alex has 53 dollars in total.

**Formal Statement and Proof:**

```
def calculate_money (tens : ℕ) (ones : ℕ) : ℕ := tens * 10 + ones * 1

example :  calculate_money 5 3 = 53 :=

begin

  rw calculate_money,

  refl,

end
```

---

**miniF2F Case**

```
theorem algebra_sqineq_unitcircatbpamblt1

(a b: ℝ)

(h₀ :  a² + b² = 1) :

a * b + (a − b) le 1 :=

begin

  nlinarith [sq_nonneg (a − b)],

end
```

---

Table 30: Nearest data points between MUSTARDSAUCE and mathlib.

**MUSTARDSAUCE v.s. mathlib (cosine similarity = -0.0361)**

**MUSTARDSAUCE Case**

**Informal Statement:**

A cube has a side length of 5 cm. What is the volume of the cube?

**Informal Proof:**

The volume of a cube is calculated by raising the side length to the power of 3. So in this case, the volume is 5 cm * 5 cm * 5 cm = 125 cubic centimeters.

**Formal Statement and Proof:**

```
def cube_volume (side_length : ℕ) : ℕ := side_length * side_length *
side_length

#eval cube_volume 5 -- returns 125
```

**mathlib Case**

**GOAL** $\alpha$: Type u, $\beta$: Type v, $\gamma$ : Type w, f : $\alpha \to \beta \to \gamma$, $l_1$ : list $\alpha \vdash \forall (l_2$ : list $\beta)$, (list.map$_2$ f $l_1$ $l_2$).length = linear_order.min $l_1$.length $l_2$.length

**PROOFSTEP** induction $l_1$; intro $l_2$; cases $l_2$; simp [*, add_one, min_succ_succ, nat.zero_min, nat.min_zero]

Table 31: Nearest data points between MUSTARDSAUCE and GSM8K.

---

**MUSTARDSAUCE v.s. GSM8K (cosine similarity = 0.5975)**

---

**MUSTARDSAUCE Case**

**Informal Statement:**

Given two vectors u = (u1,u2) and v = (v1,v2) in R$^2$, prove that the operation of vector addition, defined as u + v = (u1+v1, u2+v2), is commutative. That is, prove that for all u and v in R$^2$, u + v = v + u.

**Informal Proof:**

We can prove this by direct computation. Given the vectors u = (u1, u2) and v = (v1, v2), we have:

u + v = (u1 + v1, u2 + v2)

and

v + u = (v1 + u1, v2 + u2).

Since addition is commutative in R (the set of all real numbers), we have u1 + v1 = v1 + u1 and u2 + v2 = v2 + u2. Thus, we conclude that (u1 + v1, u2 + v2) = (v1 + u1, v2 + u2), which means that u + v = v + u.

**Formal Statement and Proof:**

```
import data.real.basic

variables (u1 u2 v1 v2 :  ℝ)

-- defining vectors as pair of real numbers

def vector := ℝ × ℝ

-- defining vector addition

def vadd (u v :  vector) :  vector := (u.1 + v.1, u.2 + v.2)

-- defining vectors u and v

def u :  vector := (u1, u2)

def v :  vector := (v1, v2)

-- commutativity of vector addition

theorem vadd_comm :  vadd (u u1 u2) (v v1 v2) = vadd (v v1 v2) (u u1 u2)
:=

begin
 unfold vadd,
 unfold u,
 unfold v,
 rw add_comm u1 v1,
 rw add_comm u2 v2,
end
```

---

**GSM8K Case**

**Question:** The local firefighters are doing a "fill the boot" fundraiser. Their goal is to raise $6300. After the first 3 hours, they have raised $2100. For how many hours do they have to fundraise in total to reach their goal, assuming an equal amount raised in every hour?

**Answer:**

The fireman raise 2100 / 3 = $<<2100/3=700>>700 per hour.

They have to fundraise for 6300 / 700 = <<6300/700=9>>9 hours.

#### 9

---

Table 32: Nearest data points between MUSTARDSAUCE and MATH.

---

**MUSTARDSAUCE v.s. MATH (cosine similarity = 0.6064)**

---

**MUSTARDSAUCE Case**

**Informal Statement:**

Given two vectors u = (u1,u2) and v = (v1,v2) in $R^2$, prove that the operation of vector addition, defined as u + v = (u1+v1, u2+v2), is commutative. That is, prove that for all u and v in $R^2$, u + v = v + u.

**Informal Proof:**

We can prove this by direct computation. Given the vectors u = (u1, u2) and v = (v1, v2), we have:

u + v = (u1 + v1, u2 + v2)

and

v + u = (v1 + u1, v2 + u2).

Since addition is commutative in R (the set of all real numbers), we have u1 + v1 = v1 + u1 and u2 + v2 = v2 + u2. Thus, we conclude that (u1 + v1, u2 + v2) = (v1 + u1, v2 + u2), which means that u + v = v + u.

**Formal Statement and Proof:**

```
import data.real.basic

variables (u1 u2 v1 v2 :  ℝ)

-- defining vectors as pair of real numbers

def vector := ℝ × ℝ

-- defining vector addition

def vadd (u v :  vector) :  vector := (u.1 + v.1, u.2 + v.2)

-- defining vectors u and v

def u :  vector := (u1, u2)

def v :  vector := (v1, v2)

-- commutativity of vector addition

theorem vadd_comm :  vadd (u u1 u2) (v v1 v2) = vadd (v v1 v2) (u u1 u2)
:=

begin
 unfold vadd,
 unfold u,
 unfold v,
 rw add_comm u1 v1,
 rw add_comm u2 v2,
end
```

---

**MATH Case**

**Problem:** If a snack-size tin of peaches has 40 calories and is 2% of a person's daily caloric requirement, how many calories fulfill a person's daily caloric requirement?

**Solution:** If 40 calories is equal to $2\% = \frac{2}{100} = \frac{1}{50}$ of a person's daily requirement, then a person's daily caloric requirement is: $4 \cdot 50 = \boxed{2000}$

**Answer**: 2000

---

