# OpenReview forum: "MUSTARD: Mastering Uniform Synthesis of Theorem and Proof Data"
_ICLR.cc/2024/Conference — ICLR 2024 spotlight_

### Official Review · Reviewer_Mb77 · 2023-10-21

**Soundness:** 4 excellent
**Presentation:** 3 good
**Contribution:** 3 good
**Rating:** 8
**Confidence:** 4

**Summary:**

This paper proposes a novel method, Mustard, for the synthesis of informal and formal theorems and proofs. MUSTARD is then applied to generate MustardSauce, a dataset of seven thousand mathematical datapoints in subjects ranging from elementary school to undergraduate curricula. The authors demonstrated that language models can improve their informal and formal reasoning abilities by fine-tuning on this large and diverse dataset.

**Strengths:**

- The proposed pipeline that synthesises informal and formal theorems and proofs is conceptually simple.
- The resulting dataset is diverse and contains step-by-step reasoning processes, which are very useful for chain-of-thought reasoning.
- There is a thorough analysis of the dataset which helps the understanding.
- It is validated that there are considerable gains to be made by fine-tuning language models on this dataset.
- The formalising step has validated effectiveness since by a simple filtering of the valid formal statements and proofs, language models can have better performance.

**Weaknesses:**

- Generating data from a language model always has the risk of data contamination. It would improve the confidence in the conclusions if it can be shown that no serious data contamination took place.

**Questions:**

What is the "valid" rate of the generated formal statements and proofs? I can't seem to find it in the paper.

---

> ### Author Response · Authors · 2023-11-18
> **Response to Reviewer Mb77**
>
> **R3Q1. The risk of data contamination.**
>
> Thank you for the valuable question.
> We check cross-contamination between MUSTARDSAUCE and the evaluation datasets inspired by the GPT-4 Technical Report [1].
> However, instead of using a substring match that may result in false negatives and false positives, we compute cosine similarities based on text-embedding-ada-002 [3] and then inspect the nearest data points in the paired datasets.
> The automated theorem proving (ATP) dataset miniF2F only contains formal statements and proofs, while the math word problem (MWP) dataset GSM8K only contains informal statements and proofs.
> For a more detailed inspection, we check the corresponding fractions in MUSTARDSAUCE.
> We demonstrate the inspected cases as follows.
> The nearest data points are dissimilar.
> Therefore, we exclude data contamination of the generated MUSTARDSAUCE according to these observations.
>
> [1] OpenAI. GPT-4 technical report. CoRR, abs/2303.08774, 2023.
>
> [3] Greene, R., Sanders, T., Weng, L., & Neelakantan, A. (2022). New and improved embedding model. https://openai.com/blog/new-and-improved-embedding-model
>
>
>
> > **MUSTARDSAUCE v.s. miniF2F (cosine similarity = 0.6439)**
> >
> > **MUSTARDSAUCE Case**
> >
> > Informal Statement:
> > Alex has 5 ten-dollar bills and 3 one-dollar bills. How much money does Alex have in total?
> >
> > Informal Proof:
> > To find out how much money Alex has in total, we need to multiply the number of each type of bill by its value. So, Alex has 5 ten-dollar bills, which equals 5 * 10 = 50 dollars. He also has 3 one-dollar bills, which equals 3 * 1 = 3 dollars. Adding these two amounts together gives 50 + 3 = 53 dollars. Therefore, Alex has 53 dollars in total.
> >
> > Formal Statement and Proof:
> > ```
> > def calculate_money (tens : ℕ) (ones : ℕ) : ℕ := tens * 10 + ones * 1
> >
> > example : calculate_money 5 3 = 53 :=
> > begin
> >   rw calculate_money,
> >   refl,
> > end
> > ```
> >
> >
> > **miniF2F Case**
> > ```
> > theorem algebra_sqineq_unitcircatbpamblt1
> >   (a b: ℝ)
> >   (h₀ : a^2 + b^2 = 1) :
> >   a * b + (a - b) ≤ 1 :=
> > begin
> >   nlinarith [sq_nonneg (a - b)],
> > end
> > ```
>
>
>
> > **MUSTARDSAUCE v.s. GSM8K (cosine similarity = 0.5975)**
> >
> > **MUSTARDSAUCE Case**
> >
> > Informal Statement:
> > Given two vectors u = (u1,u2) and v = (v1,v2) in R^2, prove that the operation of vector addition, defined as u + v = (u1+v1, u2+v2), is commutative. That is, prove that for all u and v in R^2, u + v = v + u.
> >
> > Informal Proof:
> > We can prove this by direct computation. Given the vectors u = (u1, u2) and v = (v1, v2), we have:
> > u + v = (u1 + v1, u2 + v2)
> > and
> > v + u = (v1 + u1, v2 + u2).
> > Since addition is commutative in R (the set of all real numbers), we have u1 + v1 = v1 + u1 and u2 + v2 = v2 + u2. Thus, we conclude that (u1 + v1, u2 + v2) = (v1 + u1, v2 + u2), which means that u + v = v + u.
> >
> > Formal Statement and Proof:
> > ```
> > import data.real.basic
> >
> > variables (u1 u2 v1 v2 : ℝ)
> >
> > -- defining vectors as pair of real numbers
> > def vector := ℝ × ℝ
> > -- defining vector addition
> > def vadd (u v : vector) : vector := (u.1 + v.1, u.2 + v.2)
> >
> > -- defining vectors u and v
> > def u : vector := (u1, u2)
> > def v : vector := (v1, v2)
> >
> > -- commutativity of vector addition
> > theorem vadd_comm : vadd (u u1 u2) (v v1 v2) = vadd (v v1 v2) (u u1 u2) :=
> > begin
> >   unfold vadd,
> >   unfold u,
> >   unfold v,
> >   rw add_comm u1 v1,
> >   rw add_comm u2 v2,
> > end
> > ```
> > **GSM8K Case**
> >
> > Question: The local firefighters are doing a "fill the boot" fundraiser. Their goal is to raise \\$6300.
> > After the first 3 hours, they have raised
> > \\$2100. For how many hours do they have to fundraise in total to reach their goal, assuming an equal amount raised in every hour?
> >
> > Answer:
> > The fireman raise 2100 / 3 = \\$<<2100/3=700>>700 per hour.
> > They have to fundraise for 6300 / 700 = <<6300/700=9>>9 hours.
> >
> > \#\#\#\# 9
>
>
>
>
>
> **R3Q2. What is the "valid" rate of the generated formal statements and proofs?**
>
> Thank you for the question. The "valid" rate, or the "pass rate", is the proportion of generated data (obtained from the 2nd stage of MUSTARD: Proof Generation) that are validated by the Lean Prover (obtained from the 3rd stage of MUSTARD: Proof Filtering).

---

> > ### Author Response · Authors · 2023-11-19
> > **More Contamination Check Results**
> >
> > The nearest data points between MUSTARDSAUCE and mathlib/MATH are demonstrated as follows, which are also dissimilar. Therefore, we exclude data contamination of the generated MUSTARDSAUCE according to these observations.
> >
> > > **MUSTARDSAUCE v.s. mathlib (cosine similarity = -0.0361)**
> > >
> > > **MUSTARDSAUCE case**
> > >
> > > Informal Statement:
> > > A cube has a side length of 5 cm. What is the volume of the cube?
> > >
> > > Informal Proof:
> > > The volume of a cube is calculated by raising the side length to the power of 3. So in this case, the volume is 5 cm * 5 cm * 5 cm = 125 cubic centimeters.
> > >
> > > Formal Statement and Proof:
> > > ```
> > > def cube_volume (side_length : ℕ) : ℕ := side_length * side_length * side_length
> > > #eval cube_volume 5  -- returns 125
> > > ```
> > >
> > > **mathlib Case**
> > >
> > > GOAL α : Type u, β : Type v, γ : Type w, f : α → β → γ, l₁ : list α ⊢ ∀ (l₂ : list β), (list.map₂ f l₁ l₂). length = linear_order.min l₁.length l₂.length
> > >
> > > PROOFSTEP { induction l₁; intro l₂; cases l₂; simp [*, add_one, min_succ_succ, nat.zero_min, nat.min_zero] }
> > >
> >
> >
> >
> >
> > > **MUSTARDSAUCE v.s. MATH (cosine similarity = 0.6064)**
> > >
> > > **MUSTARDSAUCE case**
> > >
> > > Informal Statement:
> > > Given two vectors u = (u1,u2) and v = (v1,v2) in R^2, prove that the operation of vector addition, defined as u + v = (u1+v1, u2+v2), is commutative. That is, prove that for all u and v in R^2, u + v = v + u.
> > >
> > > Informal Proof:
> > > We can prove this by direct computation. Given the vectors u = (u1, u2) and v = (v1, v2), we have:
> > > u + v = (u1 + v1, u2 + v2)
> > > and
> > > v + u = (v1 + u1, v2 + u2).
> > > Since addition is commutative in R (the set of all real numbers), we have u1 + v1 = v1 + u1 and u2 + v2 = v2 + u2. Thus, we conclude that (u1 + v1, u2 + v2) = (v1 + u1, v2 + u2), which means that u + v = v + u.
> > >
> > > Formal Statement and Proof:
> > > ```
> > > import data.real.basic
> > >
> > > variables (u1 u2 v1 v2 : ℝ)
> > >
> > > -- defining vectors as pair of real numbers
> > > def vector := ℝ × ℝ
> > > -- defining vector addition
> > > def vadd (u v : vector) : vector := (u.1 + v.1, u.2 + v.2)
> > >
> > > -- defining vectors u and v
> > > def u : vector := (u1, u2)
> > > def v : vector := (v1, v2)
> > >
> > > -- commutativity of vector addition
> > > theorem vadd_comm : vadd (u u1 u2) (v v1 v2) = vadd (v v1 v2) (u u1 u2) :=
> > > begin
> > >   unfold vadd,
> > >   unfold u,
> > >   unfold v,
> > >   rw add_comm u1 v1,
> > >   rw add_comm u2 v2,
> > > end
> > > ```
> > >
> > > **MATH case**
> > >
> > > Problem: If a snack-size tin of peaches has $40$ calories and is $2\%$ of a person's daily caloric requirement, how many calories fulfill a person's daily caloric requirement?
> > >
> > > Solution: If 40 calories is equal to $2\%=\frac{2}{100}=\frac{1}{50}$ of a person's daily requirement, then a person's daily caloric requirement is: $$4\cdot 50=\boxed{2000}$$
> > > Answer: 2000

---

> > > ### Comment · Reviewer_Mb77 · 2023-11-19
> > >
> > > Thank you for the data contamination analysis. My confidence in the soundness improves.

---

> > > > ### Author Response · Authors · 2023-11-20
> > > > **Many Thanks for the Positive Feedback**
> > > >
> > > > We sincerely thank Reviewer Mb77 for the positive feedback of the data contamination analysis. Thank you again for your time and effort of reviewing this paper!

---

### Official Review · Reviewer_9Wae · 2023-10-27

**Soundness:** 3 good
**Presentation:** 3 good
**Contribution:** 2 fair
**Rating:** 6
**Confidence:** 4

**Summary:**

This paper presents a framework called MUSTARD that can automatically generate natural language mathematical problems, reasoning steps, and corresponding formalized language (Lean) versions through the interaction of LLM and theorem provers. The experimental results show that the 7,335 data points generated by MUSTARD can improve the performance of Llama 2-7B and GPT2 on the GSM8k and Mathlib datasets.

**Strengths:**

- The topic of this paper is of great significance to the mathematical reasoning of large language models, and how to produce high-quality reasoning step data at low cost is indeed one of the directions the community is working towards. This paper offers certain reference value for this field.
- The writing of the paper is enjoyable, featuring a clear method description, and the experiment is easy to follow as well.
- Technically, the paper presents several innovations. For example, this paper combines natural language generation and formalized language verification, using theorem provers to verify whether the generated natural language reasoning steps are valid.
- The experimental data in this paper effectively demonstrate the value of the data generated by this framework. The evaluation section analyzes two models, multiple fine-tuning strategies, and training data, showcasing rich results.

**Weaknesses:**

The paper has some limitations:
- The improvement of involving the theorem prover is relatively minor (e.g., see $\texttt{in}$, $\texttt{ra}$, and $\texttt{va}$ in Table 4), and it raises the question of whether the best result should be derived from direct fine-tuning without any filtering and sampling.
- The subset sampled for manual verification is small (only 88 data points). Given the variance in the dataset due to the different performance of the theorem prover on various types of math problems, the results on such a small dataset may not accurately represent the statistical characteristics of the whole dataset.

Additionally, there are some minor issues in the paper:
- It is unclear to me which model is used for generating the MUSTARD dataset. It would be helpful if the authors provided the performance of the LLM used for data generation as a reference.
- The scalability of the framework is still unknown. Only 7,335 data points are generated, which is a small token size for LLM fine-tuning, and the results after fine-tuning are mediocre. Furthermore, the effectiveness of this framework for larger LLM models (e.g., 34B or 70B models or GPT3 models) remains to be investigated.
- Since both natural language problem-solving and autoformalization are generated by LLM, and only ATP is used to verify the entire scheme, it is unclear how the authors handle inconsistencies between the final result calculated by the #reduce statement (as shown in Figure 2) and the natural language. The filtering strategy in Table 13 does not cover this situation (as ATP will not report an error).
- There seems to be a significant misalignment between the generated 7,335 data points and the test set. For instance, the paper showcases a problem about the convergence of alternating series, which is within the scope of undergraduate mathematics, while the test set GSM8k only covers elementary school-level problems. Additionally, the paper mentions using the #reduce statement in Lean, which is an approach for math word problems but is not consistent with the style of various theorem proofs in Mathlib.

**Questions:**

1. Which LLM is used for data generation?

2. Does the method remain more effective when compared to fine-tuning the model using the entire generated dataset?

2. Would the method potentially be more effective if more data points were generated and a larger model was employed for fine-tuning?

---

> ### Author Response · Authors · 2023-11-18
> **Response to Reviewer 9Wae**
>
> We sincerely thank the reviewer for all the instructive comments and valuable suggestions. We have added the following experiments to address the corresponding questions.
>
> 1. We have further conducted experiments with (1) fine-tuning in the entire generated dataset (denoted as "total", <tt>tt</tt> for short), (2) fine-tuning larger Llama 2-70B, (3) evaluation on another challenging dataset MATH [2], (4) ablation study of data scalability by fine-tuning models with different ratios of data,
> to further study the benefits of the proposed framework. We have the following findings:
>
>     1) In general, fine-tuning with MUSTARDSAUCE-valid is superior to fine-tuning with MUSTARDSAUCE-invalid, MUSTARDSAUCE-random, and also the entire generated dataset. This indicates the benefits of involving formal validation in our framework.
>
>     2) The performance gaps between MUSTARDSAUCE-valid and MUSTARDSAUCE-random are generally significant. The average performance gain in automated theorem proving (Table 5) is 18.15\%, and in math word problems (Table 4) is 6.78\%. The results on the MATH dataset indicate that fine-tuning with MUSTARDSAUCE-valid is considerably beneficial. For example, GPT2-large fine-tuned with MUSTARDSAUCE-valid set has a 40\% performance gain in a zero-shot manner, and the performance of Llama 2-7B in a few-shot manner increases by 11.76\%.
>
>     3) Fine-tuning the larger Llama 2-70B shows greater performance improvements, which demonstrates the scalability of this framework.
>
>     4) The model performances increase as the generated data scales up. Therefore we expect further performance improvements when more high-quality data are included.
>
>     In summary, the extended experiments demonstrate that the proposed method generates high-quality data that are beneficial to improving language models' mathematical reasoning performance. The method is scalable to larger fine-tuned LMs and more generated data points and maintains its effectiveness.
>
> 2. We have expanded the sampled subset for manual verification from 88 data points to 200 data points.
>
> 3. The LLM we used for generating the MUSTARDSAUCE dataset is GPT-4 [1] and we have added the details in our revised manuscript.
>
> [1] OpenAI. GPT-4 technical report. CoRR, abs/2303.08774, 2023.
>
> [2] Dan Hendrycks, Collin Burns, Saurav Kadavath, Akul Arora, Steven Basart, Eric Tang, Dawn Song, and Jacob Steinhardt. Measuring mathematical problem solving with the MATH dataset. In Joaquin Vanschoren and Sai-Kit Yeung (eds.), Proceedings of the Neural Information Processing Systems Track on Datasets and Benchmarks 1, NeurIPS Datasets and Benchmarks 2021, December 2021, virtual, 2021.
>
> Please check our detailed results and response in the following.

---

> > ### Comment · Reviewer_9Wae · 2023-11-20
> > **Reply**
> >
> > Thank you for the response. I do believe that these extensive experimental results can greatly enhance the completeness and soundness of the paper. Therefore, I have increased the score of the paper to 6.

---

> > > ### Author Response · Authors · 2023-11-20
> > > **Sincere Appreciation and Some Follow-up Results**
> > >
> > > We sincerely appreciate Reviewer 9Wae for the positive recognition and for improving the scores!
> > >
> > > We have also tested Llama 2-70B on the MATH dataset. The experimental results are added to Table 8 as follows and also added to Table 28 in the main paper.
> > > Llama 2-70B fine-tuned with MUSTARDSAUCE-valid consistently outperforms the model fine-tuned with MUSTARDSAUCE-random by 8.33\% in the zero-shot manner and 5.30\% in the few-shot manner. It also surpasses the model fine-tuned with the invalid subset and the entire generated dataset.
> > > The results also suggest the effectiveness of the framework with a larger fine-tuned LM.
> > >
> > >
> > > Table 8: Compared performances on GSM8K and MATH between Llama 2-7B and Llama 2-70B.
> > >
> > > | MODEL            | Zero (G)                                       | Few (G)                                        | Zero (M)                                       | Few (M)                                         |
> > > | ---------------- | ---------------------------------------------- | ---------------------------------------------- | ---------------------------------------------- | ----------------------------------------------- |
> > > | *Baselines*      |                                                |                                                |                                                |                                                 |
> > > | Llama 2-7B       | 7.2                                            | 12.8                                           | 2.0                                            | 2.6                                             |
> > > | Llama 2-70B      | 31.7                                           | 54.1                                           | 8.8                                            | 13.4                                            |
> > > | *Fine-tuning*    |                                                |                                                |                                                |                                                 |
> > > | Llama 2-7B > tt  | 9.0                                            | 15.5                                           | 3.0                                            | 3.4                                             |
> > > | Llama 2-7B > in  | 8.3                                            | 14.4                                           | 2.4                                            | 3.2                                             |
> > > | Llama 2-7B > ra  | 8.9                                            | 14.9                                           | 2.8                                            | 3.4                                             |
> > > | Llama 2-7B > va  | 9.1 (+2.25%)  | 15.7 (+5.37%) | 3.0 (+7.14%)  | 3.8  (+11.76%) |
> > > | Llama 2-70B > tt | 36.6                                           | 55.8                                           | 10.0                                           | 14.4                                            |
> > > | Llama 2-70B > in | 33.4                                           | 53.7                                           | 9.2                                            | 13.6                                            |
> > > | Llama 2-70B > ra | 36.1                                           | 55.4                                           | 9.6                                            | 14.2                                            |
> > > | Llama 2-70B > va | 39.5 (+9.42%) | 59.1 (+6.68%) | 10.4 (+8.33%) | 15.0  (+5.30%)  |

---

> ### Author Response · Authors · 2023-11-18
> **Updated Experimental Results: Table 4**
>
> \
> Table 4: Maj1@1 results on GSM8K (G) and MATH (M). Zero: Zero-shot. Few: Few-shot. > denotes a fine-tuning step. in: MUSTARDSAUCE-invalid. ra: MUSTARDSAUCE-random. va: MUSTARDSAUCE-valid. tt: The entire generated data.
> gt: GSM8K training split.
> The updated results are ***bold and italic***.
> The values in parentheses are performance gaps between fine-tuning with MUSTARDSAUCE-valid and MUSTARDSAUCE-random.
>
> | MODEL                   | Zero (G)     | Few (G)       | Zero (M)      | Few (M)       | MODEL                 | Zero (G)      | Few (G)       | Zero (M)      | Few (M)       |
> | :---------------        | :----------- | :------------ | :------------ | :------------ | :-------------------- | :------------ | :------------ | :------------ | :------------ |
> | *Baselines*             |              |               |               |               |                       |               |               |               |               |
> | GPT2-large              | 3.4          | 5.1           | ***0.6***           | ***1.0***           | GPT2-large > gt       | 14.6          | 17.4          | ***4.6***           | ***6.8***           |
> | Llama 2-7B              | 7.2          | 12.8          | ***2.0***           | ***2.6***           | Llama 2-7B > gt       | 24.5          | 28.2          | ***10.4***          | ***12.6***          |
> | *Fine-tuning*           |              |               |               |               |                       |               |               |               |               |
> | ***GPT2-large > tt***   | ***3.9***    | ***6.3***     | ***1.2***     | ***2.0***     | ***GPT2-large > tt > gt***  | ***15.9***          | ***18.5***          | ***5.0***           | ***7.8***           |
> | GPT2-large > in         | 3.6          | 5.8           | ***1.0***           | ***1.8***           | GPT2-large > in > gt  | 14.6          | 17.9          | ***4.8***           | ***7.4***           |
> | GPT2-large > ra         | 3.8          | 6.3           | ***1.0***           | ***2.0***           | GPT2-large > ra > gt  | 15.8          | 18.4          | ***4.8***           | ***7.6***           |
> | GPT2-large > va         | 4.1  ***(+7.89%)*** | 6.1  ***(-3.17%)***  | ***1.4 (+40.00%)*** |   ***2.2 (+10.00%)*** | GPT2-large > va > gt  | 16.0  ***(+1.27%)*** | 18.7  ***(+1.63%)*** |  ***5.2 (+8.33%)***  |  ***7.8 (+2.63%)***  |
> | ***Llama 2-7B > tt***   | ***9.0***    | ***15.5***    | ***3.0***     | ***3.4***     | ***Llama 2-7B > tt > gt***  | ***26.1***           | ***30.2***           | ***11.8***          | ***13.8***          |
> | Llama 2-7B > in         | 8.3          | 14.4          | ***2.4***           | ***3.2***           | Llama 2-7B > in > gt  | 25.4          | 28.2          | ***10.8***          | ***12.8***          |
> | Llama 2-7B > ra         | 8.9          | 14.9          | ***2.8***           | ***3.4***           | Llama 2-7B > ra > gt  | 26.1          | 29.9          | ***11.6***          | ***13.6***          |
> | Llama 2-7B > va         | 9.1  ***(+2.25%)*** | 15.7  ***(+5.37%)*** |  ***3.0 (+7.14%)***  |  ***3.8 (+11.76%)*** | Llama 2-7B > va > gt  | 26.3  ***(+0.77%)*** | 30.8  ***(+3.01%)*** |  ***12.2 (+5.17%)*** | ***14.2 (+4.41%)*** |

---

> > ### Author Response · Authors · 2023-11-18
> > **Updated Experimental Results: Table 5**
> >
> > \
> > Table 5: Pass@1 results on automated theorem proving tasks. > denotes a fine-tuning step. test: MUSTARDSAUCE-test. in: MUSTARDSAUCE-invalid. ra: MUSTARDSAUCE-random. va: MUSTARDSAUCE-valid. tt: The entire generated data.
> > mt: mathlib training split. Note that the reported results on MUSTARDSAUCE-test are obtained by only fine-tuning on the MUSTARDSAUCE-valid training split.
> > The updated results are ***bold and italic***.
> > The values in parentheses are performance gaps between fine-tuning with MUSTARDSAUCE-valid and MUSTARDSAUCE-random.
> >
> > | MODEL            | mathlib       | miniF2F       | test           | MODEL                 | mathlib       | miniF2F       | test           |
> > | :--------------- | :------------ | :------------ | :------------- | :-------------------- | :------------ | :------------ | :------------- |
> > | *Baselines*      |               |               |                |                       |               |               |                |
> > | GPT2-large       | 0.0           | 0.0           | 0.0            | GPT2-large > gt       | 0.0           | 0.0           | 0.0            |
> > | Llama 2-7B       | 0.0           | 0.0           | 0.0            | Llama 2-7B > gt       | 0.0           | 0.0           | 0.0            |
> > | *Fine-tuning*    |               |               |                |                       |               |               |                |
> > | GPT2-large > in  | 2.0           | 0.0           | 6.0            | GPT2-large > in > mt  | 5.9           | 2.0           | 8.2            |
> > | GPT2-large > ra  | 3.0           | 1.2           | 7.0            | GPT2-large > ra > mt  | 6.6           | 2.9           | 9.6            |
> > | GPT2-large > va  | 3.7  ***(+23.33%)*** | 1.6  ***(+33.33%)*** | 8.3  ***(+18.57%)***  | GPT2-large > va > mt  | 7.4  ***(+12.12%)*** | 3.7  ***(+27.59%)*** | 10.6  ***(+10.42%)*** |
> > | ***Llama 2-7B > tt***  | ***8.3***           | ***2.6***           | ***11.7***           | ***Llama 2-7B > tt > mt***  | ***15.1***          | ***7.0***           | ***13.6***           |
> > | Llama 2-7B > in  | 5.8           | 1.2           | 8.6            | Llama 2-7B > in > mt  | 11.6          | 5.7           | 12.6           |
> > | Llama 2-7B > ra  | 7.5           | 2.5           | 10.4           | Llama 2-7B > ra > mt  | 14.7          | 6.6           | 13.2           |
> > | Llama 2-7B > va  | 8.7  ***(+16.00%)*** | 2.9  ***(+16.00%)*** | 12.2  ***(+17.31%)*** | Llama 2-7B > va > mt  | 15.7  ***(+6.80%)*** | 7.8  ***(+18.18%)*** | 14.4  ***(+18.18%)*** |

---

> ### Author Response · Authors · 2023-11-18
> **Responses to R2Q1, R2Q2, R2Q3**
>
> **R2Q1. The improvement of involving the theorem prover is relatively minor (e.g., see in, ra, and va in Table 4), and it raises the question of whether the best result should be derived from direct fine-tuning without any filtering and sampling.**
>
> Thank you for the question.
> We have further conducted experiments with fine-tuning in the entire generated dataset (28,320 data points, denoted as “total”, tt for short).
> The results are demonstrated in the revised Tables 4 and 5.
> We also mark and color code the performance gaps between models fine-tuned with MUSTARDSAUCE-random and MUSTARDSAUCE-valid in the tables.
> We find that models fine-tuned with the entire generated data are inferior to models fine-tuned with MUSTARDSAUCE-valid.
> Although the increase in the amount of fine-tuned data makes the model perform better compared to fine-tuning on MUSTARDSAUCE-invalid and MUSTARDSAUCE-random, the model's performance still lags behind that of fine-tuning on smaller amounts but higher quality data.
> Therefore, our proposed framework that introduces the theorem prover is effective and beneficial.
>
> **R2Q2. The subset sampled for manual verification is small.**
>
> Thank you for the valuable comments. We expand the manual verification subset by randomly selecting and manually checking another 112 data points.
> The results of the 200 data points in total are demonstrated in Table 3 as follows.
> It is shown that the results are consistent with more inspected data.
> (D4) and (D6) show significant differences in accuracy between the two groups.
> Moreover, (D1) also shows significance with the inspected data scaled up.
> The results demonstrate that the validated data have a higher quality than those without a formal validation process.
>
> Table 3: Inspection dimensions and requirements in human evaluation. IS: Informal Statement. IP: Informal Proof. FS: Formal Statement. FP: Formal Proof. RT: Reasoning Type.
> Significant $p<0.005$ are marked with **bold**.
>
> | Inspection Dimension | Requirement | Valid | Invalid | *p*-value |
> | ------- | ------- | ------- | ------- | ------- |
> | (D1) IS Correctness | *Whether the informal statement is factually correct.* | 93.50 | 83.50 | **0.00167** |
> | (D2) IS Relevance | *Whether the informal statement is relevant to each seed concept.* | 87.50 | 92.50 | 0.09604 |
> | (D3) RT Classification | *Whether the informal statement is of the required question type.* |67.00 | 68.50 | 0.74903 |
> | (D4) IP Correctness | *Whether the informal proof correctly solves the informal statement.* | 88.50 | 73.50 | **0.00012** |
> | (D5) IS-FS Alignment | *Whether the informal statement and the formal statement describe the same problem and are aligned with each other.* | 74.00 | 66.50 | 0.10138 |
> | (D6) IP-FP Alignment | *Whether the informal proof and the formal proof describe the same solution and have aligned proof steps.* |72.00 | 54.00 | **0.00018** |
>
> **R2Q3. The model used for generating the MUSTARD dataset.**
>
> We use GPT-4 [1] for generating the MUSTARD dataset. We have added the details to our revised version.
>
> [1] OpenAI. GPT-4 technical report. CoRR, abs/2303.08774, 2023.

---

> ### Author Response · Authors · 2023-11-18
> **Responses to R2Q4, R2Q5, R2Q6**
>
> **R2Q4. The effectiveness of this framework for larger LLM models (e.g., 34B or 70B models or GPT3 models) remains to be investigated.**
>
> Thank you for the instructive comment. We further conduct experiments on Llama 2-70B.
> Since the Llama 2-34B and the GPT-3 parameters are not released, we are unable to experiment with these two models.
> The compared performances between Llama 2-7B and Llama 2-70B are demonstrated in Table 8 below. The results are also updated in Table 28 in Appendix F in the main paper.
> Due to time limitations, we demonstrate results on GSM8K first.
> Llama 2-70B fine-tuned with MUSTARDSAUCE-valid performs 9.42\% higher than fine-tuned with MUSTARDSAUCE-random in the zero-shot manner, and 6.68\% higher in the few-shot manner.
> The performance gains are more significant compared to the results of Llama 2-7B.
> Therefore, the proposed framework remains effective with larger fine-tuned language models.
>
> Table 8: Compared performances on GSM8K between Llama 2-7B and Llama 2-70B.
>
> | MODEL            | Zero (G)                                       | Few (G)                                        |
> | ---------------- | ---------------------------------------------- | ---------------------------------------------- |
> | *Baselines*      |                                                |                                                |
> | Llama 2-7B       | 7.2                                            | 12.8                                           |
> | Llama 2-70B      | 31.7                                           | 54.1                                           |
> | *Fine-tuning*    |                                                |                                                |
> | Llama 2-7B > tt  | 9.0                                            | 15.5                                           |
> | Llama 2-7B > in  | 8.3                                            | 14.4                                           |
> | Llama 2-7B > ra  | 8.9                                            | 14.9                                           |
> | Llama 2-7B > va  | 9.1 (+2.25%)  | 15.7 (+5.37%) |
> | Llama 2-70B > tt | 36.6                                           | 55.8                                           |
> | Llama 2-70B > in | 33.4                                           | 53.7                                           |
> | Llama 2-70B > ra | 36.1                                           | 55.4                                           |
> | Llama 2-70B > va | 39.5 (+9.42%) | 59.1 (+6.68%) |
>
>
>
> **R2Q5. Would the method potentially be more effective if more data points were generated and a larger model was employed for fine-tuning?**
>
> Thank you for the constructive comment.
> Since generating and filtering new data with GPT-4 is costly, we instead use 75%, 50%, 25%, and 0% of current MUSTARDSAUCE-valid for fine-tuning Llama 2-7b to observe the effect of data scalability.
> We evaluate the model on all datasets.
> The results are demonstrated in Table 9 as follows.
> We have also uploaded the results in Figure 3 in the main paper.
> In general, the results on all datasets increase as the fine-tuning data scales up.
> Specifically, performances on the MUSTARD-test and mathlib show the most significant growth, and there is no decreasing trend in the growth rate.
> Therefore we expect further performance improvements when more high-quality data are included.
>
>
> Table 9: Llama 2-7b performances with different data scales for fine-tuning.
>
> | Data Scale | GSM8K | MATH | mathlib | miniF2F | test |
> | :--------- | :---- | :--- | :------ | :------ | :--- |
> | 0%         | 12.8  | 2.6  | 0.0     | 0.0     | 0.0  |
> | 25%        | 13.4  | 2.8  | 3.9     | 0.9     | 4.3  |
> | 50%        | 14.2  | 3.1  | 5.3     | 1.4     | 7.5  |
> | 75%        | 14.9  | 3.6  | 7.4     | 2.0     | 9.4  |
> | 100%       | 15.7  | 3.8  | 8.7     | 2.9     | 12.2 |
>
>
>
>
> **R2Q6. Does the method remain more effective when compared to fine-tuning the model using the entire generated dataset?**
>
> Thank you for the valuable comment. We have fine-tuned GPT2-large and Llama 2-7B with the entire generation dataset (contains 28,320 data points in total), and the results are demonstrated in Table 4 and Table 5 in the lines "GPT2-large > tt" and "Llama 2-7B > tt".
> We have also updated the results and marked them with blue in Tables 4 and 5 in the main paper.
> We find that the models fine-tuned with the entire generated dataset perform better than those fine-tuned with MUSTARDSAUCE-random and MUSTARDSAUCE-invalid but are still inferior to those fine-tuned with MUSTARDSAUCE-valid.
> Therefore, more fine-tuning data improves the model performances as usual, but the smaller MUSTARDSAUCE-valid turns out to be more helpful.
> This finding again demonstrates our motivation that the validated theorem and proof data are of higher quality and are beneficial for reasoning.

---

> ### Author Response · Authors · 2023-11-18
> **Responses to R2Q7, R2Q8**
>
> **R2Q7. The inconsistency between the #reduce statements and various theorem proofs.**
>
> Thank you for the question.
> The proportion of proofs with #reduce statements in the current MUSTARDSAUCE-valid set is 19.3%.
> MUSTARD aims to provide a uniform synthesis of theorem and proof data.
> Therefore, we have further filtered out the data points with the #reduce statements, which results in 5,922 data points.
>
>
>
> **R2Q8. The test set GSM8K only covers elementary school-level problems.**
>
> Thank you for the valuable question. We have added experiments on the MATH dataset [2] that contain more challenging math word problems, and the results are demonstrated in Table 4 as shown above.
> Both GPT2-large and Llama 2-7B show significant improvements after fine-tuning with MUSTARDSAUCE-valid. For example, "GPT2-large > va" performs 40\% higher than "GPT2-large > ra" in the zero-shot manner. And "Llama 2-7B > va" has an 11.76\% increase compared to "Llama 2-7B > ra".
> The observations further demonstrate the effectiveness of generated and validated MUSTARDSAUCE on math word problems.
>
> [2] Dan Hendrycks, Collin Burns, Saurav Kadavath, Akul Arora, Steven Basart, Eric Tang, Dawn Song, and Jacob Steinhardt. Measuring mathematical problem solving with the MATH dataset. In Joaquin Vanschoren and Sai-Kit Yeung (eds.), Proceedings of the Neural Information Processing Systems Track on Datasets and Benchmarks 1, NeurIPS Datasets and Benchmarks 2021, December 2021, virtual, 2021.

---

> ### Author Response · Authors · 2023-11-22
> **The Contributions of This Paper**
>
> Dear Reviewer 9Wae,
>
> We sincerely thank you again for your time and effort in reviewing this paper! We also thank you for your positive recognition of our extensive experiments and our contributions, as we quote as follows: "The topic of this paper is of great significance to the mathematical reasoning of large language models, ... This paper offers certain reference value for this field." "Technically, the paper presents several innovations." "The experimental data in this paper effectively demonstrate the value of the data generated by this framework."
>
> Our paper proposes a novel framework MUSTARD that synthesizes high-quality theorem and proof data and further extensively facilitates language models' mathematical reasoning capabilities.
> * The human evaluation results on 200 randomly selected data points indicate that the formal validation in MUSTARD makes a remarkable contribution to the data quality (Table 3).
> * The extensive experimental results with GPT2-large, Llama 2-7B, and Llama 2-70B on the two word math problem benchmarks (GSM8K and MATH, demonstrated in Table 4) and two automated theorem proving benchmarks (mathlib and miniF2F, shown in Table 5) demonstrate that the generated data significantly facilitates the smaller language models to perform more accurate mathematical reasoning.
> * Moreover, the ablation study on data scalability shows consistent performance increases when more data from MUSTARD are introduced, suggesting a great potential for scalability (Table 9 in the previous response and Figure 3 in the revised main paper). Fortunately, MUSTARD reduces the cost of acquiring such high-quality step-by-step complex reasoning data and obtains correct, scalable, and reusable data. Therefore, in future work, we would love to build a community in which all members can join the data synthesis process, and acquire and share more high-quality data with the whole community.

---

### Official Review · Reviewer_LURb · 2023-11-01

**Soundness:** 3 good
**Presentation:** 3 good
**Contribution:** 3 good
**Rating:** 8
**Confidence:** 4

**Summary:**

In this submission, the authors consider formal theorem proving, especially the data generation for this task. They present an approach to generate theorem and proof data. Their approach, which they call mustard performs two steps: given a "concept" such as "basic geometry", problems in natural language are generated with LLMs. In step 2 an informal proof in natural langauge is generated which is then passed iteratively into a formal theorem prover, namely lean, to generate a formal proof by refining.
This results in a theorem + proof dataset/benchmark called mustardsauce with 7335 examples including natural language problem statements and proofs and their corresponding formal proof.
Additionally, the authors provide experiments on the quality of the benchmark; where especially the experiment on fine-tuning on mustardsauce is interesting, as it significantly improves the reasoning capabilities even of small open-source models significantly.

**Strengths:**

- Interesting and relevant research area
- Experiments on the smaller models are interesting baseline for research and academic community
- The novel dataset/benchmark is valuable
- Expert user study is interesting and often not done in this field; interesting experiment that validates the quality of the benchmark

**Weaknesses:**

- Since the informal data is generated by an off-the-shelf LLM, it is unclear how this approach could be scaled to the best performing models.
- It was also unclear to me which LLM is used for the informal data generation

**Questions:**

I thank the authors for their submission!
- Could you elaborate on the low value of D3 in Table 3, the sentence on that in the paper was unclear to me?
- Which LLM is used for the informal data generation?
- If the LLM used is GPT-4 or another large off-the-shelf model, what would be the argument for this approach to help in performance of the best models? (if I misunderstood something, please correct me / give a short clarification)

---

> ### Author Response · Authors · 2023-11-18
> **Response to Reviewer LURb**
>
> **R1Q1. The LLM we used for the informal data generation.**
>
> We use GPT-4 [1] for generating both the informal and formal data.
>
> [1] OpenAI. GPT-4 technical report. CoRR, abs/2303.08774, 2023.
>
>
> **R1Q2. How this approach could be scaled to the best performing models.**
>
> Thank you for the question.
> The proposed MUSTARD contains two language models: (1) The large language model (hereinafter the LLM) that is prompt for generating the theorem and proof data, and specifically we use GPT-4 [1] in our experiments; (2) The fine-tuned language model using the generated MUSTARDSAUCE dataset (hereinafter as the fine-tuned LM) for miniF2F, mathlib, and GSM8K, and in our experiments we use GPT-2 and Llama 2.
> For LLM, we have used the best-performing GPT-4.
> For fine-tuned LMs, we fine-tune a larger and better-performing model Llama 2-70B with MUSTARDSAUCE.
> The results are demonstrated in Table 8 as follows.
> In general, larger fine-tuned LMs achieve more significant performance gains.
> For example, Llama 2-70B fine-tuned with MUSTARDSAUCE-valid performs 9.42% higher than fine-tuned with MUSTARDSAUCE-invalid in the zero-shot manner, and 6.68% higher in the few-shot manner.
> The performance gains are higher compared to the results of Llama 2-7B.
> Therefore, the proposed framework remains effective when scaled to better-performing fine-tuned LMs.
>
>
> Table 8: Compared performances on GSM8K between Llama 2-7B and Llama 2-70B.
>
> | MODEL            | Zero (G)                                       | Few (G)                                        |
> | ---------------- | ---------------------------------------------- | ---------------------------------------------- |
> | *Baselines*      |                                                |                                                |
> | Llama 2-7B       | 7.2                                            | 12.8                                           |
> | Llama 2-70B      | 31.7                                           | 54.1                                           |
> | *Fine-tuning*    |                                                |                                                |
> | Llama 2-7B > tt  | 9.0                                            | 15.5                                           |
> | Llama 2-7B > in  | 8.3                                            | 14.4                                           |
> | Llama 2-7B > ra  | 8.9                                            | 14.9                                           |
> | Llama 2-7B > va  | 9.1 (+2.25%)  | 15.7 (+5.37%) |
> | Llama 2-70B > tt | 36.6                                           | 55.8                                           |
> | Llama 2-70B > in | 33.4                                           | 53.7                                           |
> | Llama 2-70B > ra | 36.1                                           | 55.4                                           |
> | Llama 2-70B > va | 39.5 (+9.42%) | 59.1 (+6.68%) |
>
>
> **R1Q3. The low value of D3 in Table 3.**
>
> As introduced in Section 3.2, MUSTARD generates both theorem-proving problems and math word problems. D3 in Table 3 inspects whether MUSTARD accurately classifies the question types during generation.
> Specifically, we investigate instances where theorem-proving problems might be misclassified as math word problems, and vice versa, and assess if validation using Lean Prover reduces such misclassifications.
> We find that the $p$-value of D3 is greater than 0.005, showing that Lean Prover validation does not significantly influence the classification.
> Furthermore, the proportions in both Group Valid and Group Invalid in D3 are relatively lower than other inspection dimensions. One of the reasons is that MUSTARD generates a diverse range of problems from the elementary to higher educational levels, and generating theorem-proving problems at the elementary school level is specifically challenging.

---

> ### Author Response · Authors · 2023-11-20
> **Many Thanks and Updated Results for R1Q2**
>
> We thank Reviewer LURb again for your time and effort of reviewing this paper!
>
> We have updated the results of the fine-tuned Llama 2-70B on the MATH dataset, which are demonstrated in Table 8 as follows and also updated to Table 28 in the main paper.
> Llama 2-70B fine-tuned with MUSTARDSAUCE-valid consistently outperforms the model fine-tuned with MUSTARDSAUCE-random by 8.33\% in the zero-shot manner and 5.30\% in the few-shot manner. It also surpasses the model fine-tuned with the invalid subset and the entire generated dataset.
> The results also suggest the effectiveness of the framework when scaled to better-performing fine-tuned LMs.
>
>
>
> Table 8: Compared performances on GSM8K and MATH between Llama 2-7B and Llama 2-70B.
>
> | MODEL            | Zero (G)                                       | Few (G)                                        | Zero (M)                                       | Few (M)                                         |
> | ---------------- | ---------------------------------------------- | ---------------------------------------------- | ---------------------------------------------- | ----------------------------------------------- |
> | *Baselines*      |                                                |                                                |                                                |                                                 |
> | Llama 2-7B       | 7.2                                            | 12.8                                           | 2.0                                            | 2.6                                             |
> | Llama 2-70B      | 31.7                                           | 54.1                                           | 8.8                                            | 13.4                                            |
> | *Fine-tuning*    |                                                |                                                |                                                |                                                 |
> | Llama 2-7B > tt  | 9.0                                            | 15.5                                           | 3.0                                            | 3.4                                             |
> | Llama 2-7B > in  | 8.3                                            | 14.4                                           | 2.4                                            | 3.2                                             |
> | Llama 2-7B > ra  | 8.9                                            | 14.9                                           | 2.8                                            | 3.4                                             |
> | Llama 2-7B > va  | 9.1 (+2.25%)  | 15.7 (+5.37%) | 3.0 (+7.14%)  | 3.8  (+11.76%) |
> | Llama 2-70B > tt | 36.6                                           | 55.8                                           | 10.0                                           | 14.4                                            |
> | Llama 2-70B > in | 33.4                                           | 53.7                                           | 9.2                                            | 13.6                                            |
> | Llama 2-70B > ra | 36.1                                           | 55.4                                           | 9.6                                            | 14.2                                            |
> | Llama 2-70B > va | 39.5 (+9.42%) | 59.1 (+6.68%) | 10.4 (+8.33%) | 15.0  (+5.30%)  |

---

> ### Author Response · Authors · 2023-11-22
> **Further Elaboration on Table 3 on the Expert User Study**
>
> Dear Reviewer LURb,
>
> We have expanded the manual verification subset by randomly selecting and manually checking another 112 data points, resulting in 200 verified data points in total, 100 of which pass the Lean Prover (Group Valid) and 100 of which do not (Group Invalid).
> The results of the 200 data points are demonstrated in Table 3 as follows.
> With the larger verified subset, we observe that the D3 value of the **Valid** group becomes on par with the **Invalid** group. Moreover, the $p$-value of D3 is greater than 0.005, which is in line with our intuition that the Lean Prover does not help the question type classification, as we explained in the previous response that MUSTARD generates a diverse range of problems from the elementary to higher educational levels, and generating theorem-proving problems at the elementary school level is specifically challenging.
>
> More importantly,
> (D4) and (D6) show significant differences in accuracy between the two groups, which is consistent with our previous observations.
> Moreover, (D1) also shows significance with the inspected data scaled up.
> The results demonstrate that the validated data have a higher quality than those without a formal validation process.
> Therefore, our proposed MUSTARD framework is beneficial for providing such higher-quality data and being helpful for LLM mathematical reasoning.
>
>
>
> Table 3: Inspection dimensions and requirements in human evaluation. IS: Informal Statement. IP: Informal Proof. FS: Formal Statement. FP: Formal Proof. RT: Reasoning Type.
> Significant $p<0.005$ are marked with **bold**.
>
> | Inspection Dimension | Requirement | Valid | Invalid | *p*-value |
> | ------- | ------- | ------- | ------- | ------- |
> | (D1) IS Correctness | *Whether the informal statement is factually correct.* | 93.50 | 83.50 | **0.00167** |
> | (D2) IS Relevance | *Whether the informal statement is relevant to each seed concept.* | 87.50 | 92.50 | 0.09604 |
> | (D3) RT Classification | *Whether the informal statement is of the required question type.* |67.00 | 68.50 | 0.74903 |
> | (D4) IP Correctness | *Whether the informal proof correctly solves the informal statement.* | 88.50 | 73.50 | **0.00012** |
> | (D5) IS-FS Alignment | *Whether the informal statement and the formal statement describe the same problem and are aligned with each other.* | 74.00 | 66.50 | 0.10138 |
> | (D6) IP-FP Alignment | *Whether the informal proof and the formal proof describe the same solution and have aligned proof steps.* |72.00 | 54.00 | **0.00018** |

---

> > ### Comment · Reviewer_LURb · 2023-11-22
> > **Thanks!**
> >
> > I thank the authors for addressing my questions and providing more experiments on Llama models. One question remains for future work: can this approach still be applied if informal data is generated by model M (which is not GPT-4) and then M will be fine-tuned with the approach (which was my initial question and might've been a misunderstanding). Nevertheless, the dataset could be an important benchmark for the neural theorem proving community in the future. I've increased my score.

---

> > > ### Author Response · Authors · 2023-11-22
> > > **Sincere Appreciation and Response to the Further Question**
> > >
> > > We sincerely appreciate your positive feedback! And we apologize for the misunderstanding of your question. This is a very interesting setup to see if the generated data by model M can benefit its own mathematical reasoning. Thank you for this valuable suggestion! We will surely add this experiment to our future study. For example, we will try this setup with Llama 2-70B and see how it works.

---

### Author Response · Authors · 2023-11-18
**General Response to All Reviewers**

We thank all the reviewers for their time, insightful suggestions, and valuable comments. We are grateful for the positive recognition of the reviewers on our motivation (9Wae), novelty (9Wae, LURb), extensive experiments (Mb77), and good results (LuRb, 9Wae, Mb77).
Reviewer 9Wae's concern may be based on some incomplete observations.
We respond to each reviewer's comments in detail below.
We have also revised the main paper according to the reviewers' suggestions. The main changes are listed as follows.

* Experimental results on fine-tuning in the entire generated dataset (denoted as “total”, tt for short) (2) evaluation on the challenging dataset MATH [2] are updated in Tables 4 and 5.
Ablation study results of fine-tuning models with different data scales are updated in Figure 3. (To Reviewer 9Wae.)
* Results of fine-tuning the larger Llama 2-70B are updated in Table 28 in Appendix F due to space limitations. (To Reviewer 9Wae and LuRb.)
* Updated human evaluation results with 200 sampled data points to Table 3. (To Reviewer 9Wae.)
* Data contamination check between the generated MUSTARDSAUCE and the evaluation datasets. The discussion is in Appendix G due to space limitation. (To Reviewer Mb77.)
* Updated the detail of using GPT-4 [1] as the LLM for generating MUSTARD in Section 3.2, Section 4.4, and Table 7. (To Reviewers LuRb, 9Wae, and Mb77.)
* We again have checked and revised the expression of the full text.

Note that we have marked all the changes in blue. We hope our response and rebuttal revision will address the reviewers' concerns.


[1] OpenAI. GPT-4 technical report. CoRR, abs/2303.08774, 2023.

[2] Dan Hendrycks, Collin Burns, Saurav Kadavath, Akul Arora, Steven Basart, Eric Tang, Dawn Song, and Jacob Steinhardt. Measuring mathematical problem solving with the MATH dataset. In Joaquin Vanschoren and Sai-Kit Yeung (eds.), Proceedings of the Neural Information Processing Systems Track on Datasets and Benchmarks 1, NeurIPS Datasets and Benchmarks 2021, December 2021, virtual, 2021.

---

### Comment · Area_Chair_iM4h · 2023-11-19
**Contraversial paper**

This is a controversial paper.  I would like to 9Wae interact with the authors.

---

### Author Response · Authors · 2023-11-23
**Summarizing the Contributions and Findings of This Paper**

Dear AC and all reviewers,

We sincerely appreciate the time and effort you have invested in reviewing our paper. Your instructive comments and valuable suggestions have been immensely helpful in enhancing the quality of our work. We are grateful that all reviewers found our updated expanded experiments (Reviewer LURb, Reviewer 9Wae), human evaluation (Reviewer 9Wae), and data contamination analysis (Reviewer Mb77) to be beneficial. The revised results and discussions have been incorporated into the main paper and are highlighted in blue.

Additionally, we would like to outline our paper's main contributions and findings as follows:

* This paper proposes a novel framework MUSTARD that can synthesize high-quality theorem and proof data and further extensively enhances language models' mathematical reasoning capabilities. By applying the framework, we collect a high-quality dataset named MUSTARDSAUCE.
* The human evaluation on 200 data points shows significance in inspection dimensions (D4), (D6), and (D1), which indicates that the formal validation in MUSTARD makes a remarkable contribution to the data quality (Table 3).
* The extensive experimental results on word math problems and automated theorem proving benchmarks demonstrate that the generated data significantly enhances the smaller language models to perform more accurate mathematical reasoning. For example, the fine-tuned Llama 2-7B achieves on average an 18.15% relative performance gain in automated theorem proving, and 6.78% in math word problems (Tables 4 and 5).
* The models fine-tuned with the validated MUSTARDSAUCE-valid subset (7,335 data points) generally outperform that fine-tuned with the entire generated dataset (28,320 data points), which further demonstrates the importance of high-quality data (Tables 4 and 5).
* The framework can scale up to fine-tuning 70B LM (specifically, Llama 2-70B) and the performance improvements are even more noticeable (Table 8 in the response and Table 28 in the revised main paper).
* The ablation study on data scalability shows consistent performance increases when more MUSTARDSAUCE data are introduced, suggesting a great potential for scalability (Table 9 in the response and Figure 3 in the revised main paper). Therefore, in future work, we would like to build a community in which all members can join the data synthesis process, and acquire and share more high-quality data with the whole community.

\
Sincerely,

Paper 2311 Authors

---

### Meta-Review · Area_Chair_iM4h · 2023-12-14

**Metareview:**

This paper is well motivated by the desire to automate open-domain mathematical reasoning --- mathematical reasoning covering all mathematical concepts.  First some background. Lean's Mathlib contains 70,000 formal definitions and 127,000 formal proofs.  Furthermore, these definitions and proofs are accompanied by natural language comments much like the computer code read by language models.  Terri Tao has recently exploited AI assistance, with language models that have presumably read Mathlib, as one would with writing code, to dramatically accelerate the human effort needed to formally very mathematics.

This system be evaluated on mathlib lemmas --- searching for a proof give a statement --- and achieves 8.7% pass@1 on Mathlib.

I believe that this paper is well above acceptance threshold.

**Justification For Why Not Higher Score:**

There is no higher score.

**Justification For Why Not Lower Score:**

I think it would be nice to represent this area in the orals at ICLR.  I'm not sure this is the best paper.

---

### Decision · Program_Chairs · 2024-01-16

Accept (spotlight)